



# Elevated sources of cobalt in the Arctic Ocean

Randelle M. Bundy[1,a,*], Alessandro Tagliabue[2], Nicholas J. Hawco[1,4], Peter L. Morton[3], Benjamin S. Twining[4], Mariko Hatta[5], Abigail Noble[1,b], Mattias R. Cape[1,a], Seth G. John[6], Jay T. Cullen[7] and Mak A. Saito[1]

[1]Department of Marine Chemistry and Geochemistry, Woods Hole Oceanographic Institution, Woods Hole, MA, USA
[2]School of Environmental Sciences, University of Liverpool, Liverpool, United Kingdom
[3]National High Magnetic Field Laboratory, Tallahassee, FL, USA
[4]Bigelow Laboratory for Ocean Sciences, East Boothbay, ME, USA
[5]Department of Oceanography, University of Hawai'i at Manoa, Honolulu, HI
[6]Department of Earth Sciences, University of Southern California, Los Angeles, CA, USA
[7]School of Earth and Ocean Sciences, University of Victoria, Victoria, BC, Canada
[a] School of Oceanography, University of Washington, Seattle, WA, USA
[b]California Department of Toxic Substances Control, Sacramento, CA, USA

*corresponding author: msaito@whoi.edu

Keywords: cobalt, GEOTRACES, Arctic Ocean, biogeochemical model

Running header: Elevated cobalt in the Arctic





## 1 Abstract

Cobalt (Co) is an important bioactive trace metal that can limit or co-limit phytoplankton growth in many regions of the ocean. Total dissolved and labile Co measurements in the Canadian sector of the Arctic Ocean during U.S. GEOTRACES Arctic expedition (GN01) and the Canadian International Polar Year-GEOTRACES expedition (GIPY14) revealed a dynamic biogeochemical cycle for Co in this basin. The major sources of Co in the Arctic were from shelf regions and rivers, with only minimal contributions from other freshwater sources (sea ice, snow) and aeolian deposition. The most striking feature was the extremely high concentrations of dissolved Co in the upper 100 m, with concentrations routinely exceeding 800 pmol L$^{-1}$ over the shelf regions. This plume of high Co persisted throughout the Arctic basin and extended to the North Pole, where sources of Co shifted from primarily shelf-derived to riverine, as freshwater from Arctic rivers was entrained in the Transpolar Drift. Dissolved Co was also strongly organically-complexed in the Arctic, ranging from 70-100% complexed in the surface and deep ocean, respectively. Deep water concentrations of dissolved Co were remarkably consistent throughout the basin (~55 pmol L$^{-1}$), with concentrations reflecting those of deep Atlantic water and deep ocean scavenging of dissolved Co. A biogeochemical model of Co cycling was used to support the hypothesis that the majority of the high surface Co in the Arctic was emanating from the shelf. The model showed that the high concentrations of Co observed along the transect were due to the large shelf area of the Arctic, as well as dampened scavenging of Co by manganese (Mn)-oxidizing bacteria due to the lower temperatures. The majority of this scavenging appears to have occurred in the upper 200 m, with minimal additional scavenging below this depth. Preliminary evidence suggests that both dissolved and labile Co are increasing over time on the Arctic shelf, and the elevated surface concentrations of Co likely leads to a net flux of Co out of the Arctic, with implications for downstream biological uptake of Co in the North Atlantic and elevated Co in North Atlantic Deep Water. Understanding the current distributions of Co in the Arctic will be important for constraining changes to Co inputs resulting from regional intensification of freshwater fluxes from ice and permafrost melt in response to ongoing climate change.





## 1. Introduction

Cobalt (Co) is an essential micronutrient in the ocean. It is utilized by eukaryotic phytoplankton as a substitute for zinc (Zn) in the metalloenzyme carbonic anhydrase (Lane and Morel, 2000; Sunda and Huntsman, 1995; Yee and Morel, 1996), and cyanobacteria have an absolute requirement for Co (Hawco and Saito, 2018; Saito et al., 2002; Sunda and Huntsman, 1995). Co is also the metal center in the micronutrient cobalamin, or vitamin $B_{12}$. In most ocean basins, dissolved Co (dCo; < 0.2 µm) is extremely scarce in surface waters (< 10 pmol $L^{-1}$), and is strongly complexed by a pool of thus far uncharacterized organic Co-binding ligands (Saito et al., 2005; Saito and Moffett, 2001). Due to its low concentrations and strong organic complexation, dCo has been found to be the limiting or co-limiting nutrient for phytoplankton growth in several regions (Bertrand et al., 2007, 2015; Browning et al., 2017; Martin et al., 1989; Moore et al., 2013; Saito et al., 2005). Growth limitation can be due to either a lack of dCo, or cobalamin (Bertrand et al., 2012; Bertrand et al., 2007; Browning et al., 2017), as cobalamin is only synthesized by cyanobacteria and some archaea (Doxey et al., 2015). However, many phytoplankton utilize cobalamin for the synthesis of methionine (Yee and Morel, 1996; Zhang et al., 2009), and therefore must obtain it from the natural environment (Heal et al., 2017).

Co is taken up as a micronutrient by phytoplankton in surface waters and is regenerated from sinking organic matter at depth, but it is also prone to intense scavenging throughout the mesopelagic ocean (Hawco et al., 2018; Saito et al., 2017). The strongest removal mechanism for dissolved Co (dCo) is through co-precipitation of dCo with manganese (Mn) by Mn-oxidizing bacteria, due to their similar redox properties and ionic radii (Cowen and Bruland, 1985; Moffett and Ho, 1996; Sunda and Huntsman, 1988). Several sources of Co to the ocean have been identified, including riverine (Tovar-Sánchez et al., 2004; Zhang et al., 1990), coastal sediments (Hawco et al., 2016; Noble et al., 2012; Noble et al., 2017), and to a lesser extent hydrothermal and aeolian inputs (Shelley et al., 2012; Thuróczy et al., 2010). The largest reservoirs of dCo thus far have been seen in oxygen deficient zones, likely due to a combination of low oxygen concentrations at the sediment-water interface and advection from reducing sediments, as well as enhanced regeneration in low oxygen waters (Hawco et al., 2016; Noble et al., 2012; Noble et al., 2017). These oxygen minimum zone sources of dCo exert an important control on the inventory of dCo, which is likely sensitive to small perturbations in bottom water oxygen concentrations (Hawco et al., 2018; Tagliabue et al., 2018).

It is important to understand the sources and sinks and internal cycling of dCo due to its important role as a micronutrient. However, Co has one of the most complex biogeochemical cycles of all of the trace metals. Thousands of measurements of both total dCo and weakly complexed and/or inorganic or "labile" Co (LCo) and particulate Co (pCo) now exist in the ocean, greatly improving our understanding of Co cycling and have facilitated the representation of the biogeochemical model of Co to be included in global ocean models (Tagliabue et al., 2018). Several observational zonal transects have been generated by large-scale programs including the international GEOTRACES program, among others. Large datasets now exist in the North Atlantic (Baars and Croot, 2015; Dulaquais et al., 2014a; Dulaquais et al., 2014b; Noble et al., 2017), South Atlantic (Noble et al., 2012), South Pacific (Hawco et al., 2016), Southern Ocean (Bown et al., 2011; Saito et al., 2010), and Mediterranean Sea (Dulaquais et al., 2017). While these studies document certain features in dCo distributions that are common in all





basins, there exist unique, regionally specific features. For example, although several of the
datasets included regions influenced by hydrothermal inputs, no significant Co feature associated
with the neutrally buoyant plume was observed (Hawco et al., 2016) and only a small point
source of dCo was observed in another case (Noble et al., 2016). Additionally, strong nepheloid
layers were shown to be a bottom water sink for dCo (Noble et al., 2016).

Although the global coverage of Co measurements has greatly improved over the last decade, no
published measurements to our knowledge have been made in the Arctic Ocean. The Arctic
Ocean is arguably the most dynamic of the ocean basins, and is changing rapidly due to warmer
temperatures affecting the maximal sea ice extent (Screen and Simmonds, 2010; Stroeve et al.,
2012), the melting of permafrost (Jorgenson et al., 2006), and additional inputs of meltwater and
river water (Johannessen et al., 2004; Serreze and Barry, 2011). The Arctic Ocean is also likely
distinct in terms of Co cycling compared to other ocean basins due to its large shelf area,
restricted circulation, and potentially distinct Co sources including sea ice, snow, and highly
seasonal riverine inputs. The Arctic Ocean is known to have high concentrations of dissolved
organic matter (DOM), which could influence the organic complexation of Co in this ocean
basin. This study examined two distinct transects of dCo, LCo and one transect of pCo in the
Canadian sector of the Arctic Ocean. We then used a Co biogeochemical model (Tagliabue et al.,
2018) in order to interpret the role of external sources and internal cycling to the observed Co
distributions, the potential of the Arctic to be a net source of Co to the North Atlantic, and to
identify Co sources and sinks that may be sensitive to future changes in this rapidly changing
ocean basin.

**2. Methods**

*2.1 Sample collection and handling*

*2.1.1. Water column samples*
Samples were collected on two expeditions in the Canadian section of the Arctic Ocean (Fig. 1).
The first set of samples (*n* = 107) were collected on board the CCGS *Amundsen* from August 27,
2009 to September 12, 2009 in the Beaufort Sea as part of the Canadian IPY-GEOTRACES
program (ArcticNet 0903; GIPY14). The second set of samples (*n* = 361) were collected on
board the USCGC *Healy* (HLY1502) on the U.S. GEOTRACES Arctic expedition (GN01) from
August 9, 2015-October 12, 2015. The Canadian GEOTRACES expedition sampled along the
shelf and slope in the Beaufort Sea. The U.S. GEOTRACES expedition sailed in and out of
Dutch Harbor, Alaska, and traversed across the Bering Shelf and Makarov Basin before reaching
the North Pole on September 5, 2015 and returning south across the Canada Basin. Samples from
the Canadian GEOTRACES expedition were collected using a trace metal rosette system fitted
with 12 x 12 L GO-FLO bottles (General Oceanics), and only the dCo and LCo samples
collected in the water column from this study are discussed here. All other metadata from this
expedition can be found at http://www.bodc.ac.uk/geotraces/data/. Samples from the U.S.
GEOTRACES expedition were collected using the U.S. GEOTRACES trace metal clean rosette
outfitted with twenty-four 12 L GO-FLO bottles and a Vectran conducting hydrowire (Cutter and
Bruland, 2012). Two GO-FLO bottles were triggered at each depth during the trace metal
hydrocasts. One bottle was used for particulate trace metal sampling, and the other was used for
all dissolved metal and macronutrient analyses. Upon recovery of the sampling system, the GO-



FLO bottles were immediately brought inside a twenty-foot ISO container van. Sampling for
bulk particulate trace metal samples has been described in detail elsewhere (Twining et al.,
2015). Briefly, samples were filtered with 25 mm Supor 0.45 µm polyethersulfone filters
mounted in Swinnex polypropylene filter holders (Twining et al., 2015). After filtering, the
volume that passed through the filter was measured and a vacuum was applied to remove any
remaining seawater on the filters. The filters were stored in trace metal clean centrifuge tubes
and frozen at -20°C until analysis (Twining et al., 2015). Dissolved trace metal and nutrient
samples were filtered with a 0.2 µm capsule filter (Acropak-200, VWR International) under
pressurized filtered air (Cutter and Bruland, 2012). Samples for dCo and LCo from the Canadian
GEOTRACES expedition were collected similarly, but were unfiltered. Nutrient samples were
analyzed immediately on-board by the Ocean Data Facility at Scripps Institution of
Oceanography. Samples for dCo were placed in two separate 60 mL Citranox-soaked (1%) and
acid-cleaned low-density polyethylene (LDPE) bottles and were filled until there was no head
space (Noble et al., 2012; Noble et al., 2017). One sample was used for LCo analyses and the
other was used for total dCo analyses.

*2.1.2 Ice hole samples*

Ice hole samples were only analyzed from the U.S. GEOTRACES cruise (GN01). Seawater from
ice holes for Co analyses was collected using Teflon coated Tygon tubing and a rotary pump
with plastic wetted parts (IWAKI magnetic drive pump, model WMD-30LFY-115) from a hole
at the station's sea ice floe. The hole was made with an ice corer (Kovacs 9 cm diameter Mak II
corer), and allowed to sit undisturbed for ~ 1 hour under a canvas tent prior to sampling. Samples
were collected from 1, 5 or 20 m at several sites. Seawater was filtered in-line with a 0.2 µm
filter (Acropak-200 capsule filter) and dispensed into a carboy, where it was homogenized and
brought back to the clean lab on board the ship. Sub-samples were taken for dCo from this
carboy, and stored as described below for other water column dissolved samples. Additional
details on ice hole samples can be found elsewhere (Marsay et al., 2018).

*2.2 Sample storage*

Total dCo and LCo samples were stored in two distinct ways. Oxygen concentrations have been
found to have a significant effect on storage of dCo samples (Noble et al., 2017). Although the
mechanism has not been fully explained, loss of some dCo species has been observed in the
presence of oxygen on both acidified and non-acidified samples across regions with active
biological gradients (Hawco et al., 2016; Noble et al., 2012; Noble et al., 2017, 2008). Since dCo
and LCo analyses were not able to be performed at sea on either expedition, groups of six dCo
samples from the U.S. expedition from a single cast were double-bagged and stored in a gas-
impermeable plastic bag (Ampac) along with 3-4 gas-absorbing satchels (Mitsubishi Gas
Chemical- model RP-3K). This outer bag was heat-sealed and samples were kept refrigerated
(4°C) until analysis (Hawco et al., 2016, 2018; Noble et al., 2016). LCo samples were double-
bagged and stored at 4°C until analysis. Samples were hand-carried at the termination of the
GN01 expedition to Woods Hole Oceanographic Institution, and all samples were analyzed
within three months. Samples from the Canadian GEOTRACES expedition (GIPY14) were
collected as unfiltered samples, and were not stored in gas-impermeable bags prior to analysis, as
the effects of oxygen on dCo loss were not known at the time of the expedition. It is possible



there could have been some loss of dCo during the time between sample collection and analyses
(approximately one year), and thus these concentrations could be underestimated. Additional
discussion on how storage may have impacted these results is discussed in section *3.2.2* and *4.3*.
*2.3 Reagent preparation*
All reagents were prepared in acid-clean plastic bottles, and prior to analyses large batches of
each reagent were made in order to have consistent reagent batches for all analyses. For dCo and
LCo analyses, a 0.5 mol L$^{-1}$ EPPS (N-(2-hydroxyethyl)piperazine-N-(3-propanesulfonic acid))
buffer and a 1.5 M NaNO$_2$ solution were prepared in Milli-Q (18 MΩ) and chelexed (Chelex-
100, Biorad) to remove trace metal contaminants. Dimethylglyoxime (DMG) was prepared by
first making a 10$^{-3}$ mol L$^{-1}$ EDTA solution in Milli-Q and adding 1.2 g of DMG. This solution
was warmed by carefully microwaving at 50% power to prevent boiling, until the DMG was
fully dissolved. The solution was placed on ice and left at 4°C to recrystallize overnight. The
supernatant was decanted, and the remaining crystals were poured into an acid-cleaned plastic
weigh boat and the remaining liquid was left to evaporate overnight in a Class-100 clean hood.
Once dry, the remaining DMG was added to an Optima methanol solution for a final
concentration of 0.1 mol L$^{-1}$ DMG. A 1.5 mol L$^{-1}$ solution of sodium nitrite was prepared by
placing sodium nitrite in Milli-Q and chelexing the solution before use to remove trace metal
contaminants. A Co standard solution was prepared weekly by adding 29.5 μl of a 1 mg L$^{-1}$ Co
AA standard (SPEX CertiPrep) to 100 mL of Milli-Q in a volumetric flask. For each new Co
standard that was prepared during sample runs, an approximately 1 mL aliquot was saved for
later analyses to ensure no variation was seen between batches. More information on reagent
preparations can be found at https://www.protocols.io/researchers/randie-bundy/publications.
*2.4 Dissolved and labile cobalt determinations*
The dCo and LCo measurements were determined using a modified cathodic stripping
voltammetry method (Saito and Moffett, 2001) for the GIPY14 samples, and a fully automated
method based on Hawco et al. (2016) for the GN01 samples (Hawco et al., 2016). Measurements
for both sample sets were performed using a Metrohm 663 VA stand connected to an Eco-
Chemie μAutolabIII system. Peak determinations for samples collected on GIPY14 were
completed as described in Noble et al. (2012). Sample automation and data acquisition for
samples from GN01 was completed using NOVA 1.8 software (Metrohm Autolab), and peak
determination was completed using a custom MATLAB code (see section 2.6).
The dCo samples were UV-irradiated for one hour in a temperature-controlled UV system prior
to analysis to remove any strong organic ligands that may prevent DMG from effectively binding
the entire dCo pool. For the GIPY14 samples, a modified temperature controlled UV system
(Metrohm 705 Digestor) was used (Hawco et al., 2016), while for GN01 samples an integrated
temperature-controlled (18°C) digestor was used (Metrohm 909 Digestor). In both cases samples
were placed in acid-cleaned and Milli-Q conditioned 15 mL quartz tubes. After irradiation, 11
mL of each sample was placed into acid-cleaned and sample-rinsed 15 mL polypropylene tubes.
For GIPY14 samples a final concentration of 353 μmol L$^{-1}$ DMG and 3 mmol L$^{-1}$ EPPS was
added to each sample before analysis (Noble et al., 2016), and for GN01 samples a final
concentration of 400 μmol L$^{-1}$ DMG and 7.6 mmol L$^{-1}$ EPPS was added to each sample before



analysis. Samples were then inverted several times before either being analyzed individually or
being placed on the autosampler (Metrohm 858 Sample Processor). For autosampler analyses,
the system was flushed with Milli-Q and 2 mL of sample were used to condition the tubing and
the Teflon analysis cup. Then 8.5 mL of sample was dosed into the cup automatically by a
Dosino 800 burette (Metrohm), along with 1.5 mL addition of 1.5 M $NaNO_2$ for a final analysis
volume of 10 mL. Samples were purged for 180 s with $N_2$ (high purity, > 99.99%) and
conditioned at -0.6 V for 90 s. The inorganic Co in the sample that was complexed by DMG
($\log K^{cond}$ = 11.5±0.3) forms a bis-complex with $Co^{2+}$ that absorbs to the hanging mercury drop
electrode (Saito and Moffett, 2001). The $Co^{2+}$ and the DMG are both reduced at the electrode
surface using a fast-linear sweep (from -0.6 V to -1.4 V at 10 V $s^{-1}$) and the height of the
$Co(DMG)_2$ reduction peak that appears at -1.15 V is proportional to the dCo concentration in the
sample. The dCo was quantified by triplicate scans of the sample, followed by four standard
additions of either 25 or 50 pmol $L^{-1}$ per addition that were dosed directly into the Teflon
analysis cup. The slope of the linear regression of these additions and triplicate "zero" scans
were used to calculate the individual sample-specific sensitivity (nA $pmol^{-1}$ $L^{-1}$). The average of
the three "zero addition" scans was then divided by the sensitivity and then corrected for the
volume of the reagent, and the blank (see section 2.5). In between sample batches, or before
analyzing LCo samples, the entire auto-sampling system was rinsed with 10% HCl and then
Milli-Q.
LCo measurements were made similarly to the dCo measurements, with the following
amendments. LCo samples were not UV-irradiated, and 400 µmol $L^{-1}$ DMG was added to 11 mL
of sample and was equilibrated for at least 8 hours (overnight) in conditioned 15 mL
polypropylene tubes. Immediately prior to placement of the sample on the autosampler, EPPS
was added and the samples were analyzed as described above for dCo analyses. LCo
measurements are thus operationally defined as the fraction of dCo that is labile to 400 µmol $L^{-1}$
DMG over the equilibration period (Hawco et al., 2016; Noble et al., 2012).
*2.5 Blanks and standards*
The blank was prepared by UV-irradiating low dCo seawater for one hour. After UV-irradiation,
the seawater was passed slowly through a Chelex-100 column to remove any metals. The clean
seawater was then UV-irradiated a second time before being analyzed.
The blank used for GIPY14 samples was analyzed at the beginning and the end of the sample
analyses to ensure the blank was consistent between runs. GEOTRACES consensus reference
materials were also analyzed along with GIPY14 samples, the results of which are reported
elsewhere (Noble et al., 2016).
For the GN01 samples, enough seawater was prepared in order to use the same blank seawater
for all of the subsequent sample analyses and the blank was analyzed regularly with each batch
of samples (every 10-20 samples). A combination of consensus reference materials and an in-
house seawater consistency standard were used throughout the sample analyses (Table 1). SAFe
and GEOTRACES standards were analyzed to ensure the accuracy of the sample measurements,
and were slowly neutralized drop wise with 1 N ammonium hydroxide (Optima, Fisher
Scientific) until reaching a pH of approximately 8. Aliquots of the SAFe and GEOTRACES



samples were then placed in conditioned quartz tubes and UV-irradiated for one hour, before
being analyzed as described above for dCo measurements. The consistency standard was
prepared by UV-irradiating 2 L of Southern Ocean trace metal clean seawater as described above
and was analyzed with each batch of samples to ensure consistency between sample runs.
*2.6 Dissolved and labile cobalt data processing*
Peak heights for the dCo and LCo samples for the GIPY14 dataset were determined in NOVA
1.8 software (Noble et al., 2016). All dCo and LCo peaks from the GN01 dataset were calculated
using custom MATLAB code available on GitHub (https://github.com/rmbundy/voltammetry).
Text files of the data output from NOVA 1.8 software were saved automatically from each scan,
and processed in MATALB to determine the dCo and LCo peak heights. The signal was
smoothed using the Savitzky-Golay smoothing function (span 5, degree 3), and the first
derivative of the voltammetric signal between -1.4 and -1.1 V was calculated in order to find the
start and end of the Co(DMG)$_2$ peak. The baseline was drawn and linearly interpolated between
the start and the end of the peak. The final peak height was determined by finding the maximum
of the signal and subtracting it from the baseline. Peak heights from the "zero addition" scans
were plotted along with the standard additions, and a linear regression was computed from all
seven scans. Data was flagged if the $r^2$ of the slope was < 0.97, and samples were re-analyzed.
*2.7 Dissolved and particulate manganese measurements*
The 0.2 μm-filtered seawater samples for dissolved manganese (dMn) were acidified to pH 2
using sub-boiling distilled HCl. The filtered subsamples were drawn into acid pre-washed 125
mL polymethylpentene bottles after three sample rinses, and the sample bottles were stored in
polyethylene bags in the dark at room temperature before analyses, which was usually within 24
h of collection. Prior to analysis, samples for manganese (dMn) were acidified by adding 125 μL
sub-boiling distilled 6 N HCl. Since the samples were determined dissolved iron (dFe) as well,
the obtained samples were then microwaved in groups of 4 for 3 min in a 900 W microwave
oven to achieve a temperature of 60±10°C in an effort to release dFe from complexation in the
samples. Samples were allowed to cool for at least 1 h prior to flow injection analysis. dMn were
determined in the filtered, acidified, microwave-treated subsamples using shipboard flow inject
ion analysis (FIA) method (Resing and Mottl, 1992). Samples were analyzed in groups of 8, and
the samples collected at each station were generally analyzed together during the same day. A 3-
minute pre-concentration of sample (~9 ml) onto an 8-hydroxyquinoline (8-HQ) resin column
yielded a detection limit of 0.55 nmol L$^{-1}$ and a precision of 1.16% at 2.7 nmol L$^{-1}$.
Particulate trace element concentrations were determined through a total digestion procedure as
described in Ohnemus et al. (2014) and Twining et al. (2015). Briefly, approximately 7 L of
contamination-free seawater were filtered directly from Teflon-coated GO-Flo sampling bottles
over acid-washed 47-mm (shelf stations) or 25-mm (open basin stations) PES Supor filters.
Filters were divided in half, and one half was digested for 3 hours at 100-120°C in sealed Teflon
vials containing 4 M HCl, 4 M HNO$_3$, and 4 M HF (Fisher Optima), which digests the marine
suspended particulate matter (SPM) but leaves the PES filter mostly intact. The PES filters were
rinsed with ultrahigh purity water (18.2 MΩ cm$^{-1}$) and removed from the digestion vials, and 60
μL of sulfuric acid (Optima) and 20 μL of hydrogen peroxide (Fisher Optima) were added to the





vials to digest any filter fragments. The digest solution was taken to dryness at ~210°C (8-24
hours). The digest residue was re-dissolved in 4 mL of 0.32 M $HNO_3$ before measuring the total
particulate Co, Mn and phosphorous (pCo, pMn, pP) concentrations by inductively coupled
plasma mass spectrometry (ICP-MS; Thermo Element 2, National High Magnetic Field
Laboratory, Tallahassee, Florida). Major and trace element concentrations were calibrated using
an external multi-element standard curve and corrected for instrument drift using a 10-ppb
indium internal standard (Twining et al., 2019).
*2.8 Biogeochemical modeling of Co in the Arctic*
Modeling runs in the Arctic Ocean were completed using a previously published biogeochemical
model for Co (Tagliabue et al., 2018). Briefly, the Co model is part of the PISCES-v2 model and
has an additional six tracers for Co, including dCo, scavenged Co (associated with Mn oxides),
Co within in diatoms, Co in nanoplankton, small particulate organic Co, and large particulate
organic Co (Tagliabue et al., 2018). The PISCES model is an excellent platform for these studies
as it has a detailed representation of ocean biogeochemical cycling and has been used for a range
of different studies. Measured pCo is equal to the sum of all of the particulate Co tracers in the
model (including living and non-living pools). Excretion of Co is also simulated in a similar
manner as iron (Fe) in PISCES-v2, with a fixed Co/C ratio in both mico- and meso-zooplankton
that sets the excretion of dCo as a function of the Co content of their food (Tagliabue et al.,
2018). The background biogeochemical model presented in Tagliabue et al. (2018) was modified
slightly for this work, most notably an improved particle flux scheme (Aumont et al., 2017), with
the Co specific parameterizations left unchanged. We used the model to assess the role of
different processes by conducting sensitivity tests whereby the sedimentary Co source was
eliminated, the riverine Co source was eliminated, the slowdown of Co scavenging at lower
oxygen was removed (meaning oxygen did not affect Co scavenging) and the change in Co
scavenging due to variations in bacterial biomass was instead set to a constant value. By
comparing the results of these four sensitivity experiments to the control model, we were able to
quantify the relative contributions of different external sources and internal cycling process.
**3. Results**
*3.1 Oceanographic context*
The Arctic Ocean is a unique ocean basin. The surface circulation in the Arctic is characterized
by a clockwise current that entrains shelf water from the Chukchi and Eurasian shelfs, before
being swept across the North Pole by the Transpolar Drift (TPD; Fig. 1). This current is
distinguished by its low salinity and elevated concentrations of dissolved organic carbon (DOC)
(Klunder et al., 2012; Wheeler et al., 1997). The Arctic Ocean is a highly stratified system, with
little mixing between the major water masses (Steele et al., 2004). The major water masses that
enter the Arctic through the Bering Strait are the upper modified Pacific water (mPW) and the
Pacific halocline water (PHW). The mPW includes inputs from the Bering shelf, as well as
freshwater inputs from rivers, sea ice melt, and glacially modified waters. PHW includes some
influences from Bering Sea water (BSW; including both summer and winter water (Steele et al.,
2004)). Atlantic water (AW) comprises the bulk of the intermediate and deep waters of the
Arctic basin. These major water masses (mPW, PHW, AW) can be distinguished from the high-





resolution nutrient, oxygen and salinity data from the conventional CTD rosette stations in the
sampling region (Fig. 2). The mPW is characteristic of low salinity (31 < S < 32) and nutrients
(Fig. 2), and contains contributions from Alaskan Coastal Water (Steele et al., 2004), as well as
other modified water masses from the shelf. The PHW can be clearly identified from the elevated
macronutrient concentrations (Fig. 2D), that extend from both shelf regions, but do not quite
reach the stations in the vicinity of the North Pole (stations 27-37). PHW can be identified both
by its elevated nutrient concentrations, as well as a temperature maximum within the salinity
range of 31-33 (Steele et al., 2004; Steele and Boyd, 1998) (Fig. 2A, C). The AW comprises a
relatively uniform deep layer throughout the entire Arctic basin. AW enters the Arctic through
the Fram Strait and Barents Sea and cycles in a cyclonic direction around the Eurasian Basin and
Canadian Basin (Aagaard and Carmack, 1989; Carmack et al., 1995) and is characterized by
higher salinities (> 33), its temperature (~ -1.0°C) and lower nutrient concentrations (silicate < 5
$\mu$mol L$^{-1}$).
*3.2 Dissolved cobalt distributions*
*3.2.1 Dissolved cobalt standards and blanks*
Blank and consensus values for the GIPY14 dataset are reported elsewhere (Noble et al., 2016).
The dCo blank for the GN01 analyses was 2.5$\pm$0.7 pmol L$^{-1}$ ($n$ = 29), with a corresponding limit
of detection of 2.1 pmol L$^{-1}$ (3 times the standard deviation of the blank; Table 1). To address
consistency between runs an internal standard was measured ($n$ = 26) and showed little variation
amongst analyses (Table 1). Several consensus standards were also analyzed yielding results that
were consistent with reported values (SAFe D1 = 47.9$\pm$2.1 ($n$ = 3), SAFe D2 = 45.2$\pm$2.1 ($n$ =
3), GSP = 2.4$\pm$1.8 ($n$ = 3), and GSC = 77.9$\pm$2.8 ($n$ = 3); Table 1).
*3.2.2 Elevated dissolved cobalt in surface waters*
The dCo profiles in the Arctic resembled a "scavenged-like" profile throughout the majority of
the transect and were distinct from recent U.S. GEOTRACES efforts in the North Atlantic
(Noble et al., 2016) and Eastern Tropical South Pacific (Hawco et al., 2016; Fig. 3). When
median dCo concentrations from this study are binned by depth, the upper 50 m in the Arctic
contains a median dCo concentration approximately 10 times higher than that of surface waters
in the North Atlantic or South Pacific (Dulaquais et al., 2014; Hawco et al., 2016; Noble et al.,
2017, 2012). Profiles in the Arctic also show no perceptible mid-depth maximum analogous to
either the Atlantic or Pacific (Fig. 3), and instead dCo concentrations rapidly decline until
reaching values of approximately 50-60 pmol L$^{-1}$. These concentrations in deep waters are
slightly lower than the deep Atlantic and closer to background Pacific levels (~30-40 pmol L$^{-1}$).
The dCo distributions were highly elevated in surface waters (< 100 m) in the shelf regions (Fig.
4A-C, P-R) and these high concentrations persisted out into the basin in the vicinity of the North
Pole (Fig. 4F-H). In the Bering Sea, dCo in surface waters ranged from 131-156 pmol L$^{-1}$ in the
upper 40 m, with an apparent surface or sub-surface minimum associated with biological
drawdown (Fig. 4A). Concentrations significantly increased in stations near the Bering Strait
(stations 2-6; Fig. 4B), where dCo reached up to 457 pmol L$^{-1}$ in surface waters (Fig. 4B; Fig. 5),
and was even higher in bottom waters, sometimes exceeding 1.5 nmol L$^{-1}$ (Fig. 4B; Fig. 5).
Surface enrichment of dCo was even more pronounced on the Chukchi shelf, where



concentrations consistently exceeded 800 pmol L$^{-1}$ (Fig. 4Q; Fig. 5). The dCo and LCo
concentrations from the Canadian GEOTRACES expedition in 2009 also had near surface
maxima in dCo and LCo, with up to 300 pmol L$^{-1}$ dCo (Fig. 4R). These concentrations were
lower than nearby samples collected in 2015 (Fig. 4P, Q), which contained up to three times
more dCo in the upper 100 m.

The elevated dCo concentrations on both shelves traversed by the U.S. expedition persisted
throughout the marginal ice zone (MIZ; stations 12-17, 51-54) and into the Canada basin
(stations 12-26), following similar patterns in dFe and dMn (L. Jensen and M. Hatta pers.
comm.). Some high concentrations of dCo were observed in the region of the MIZ and in
samples with pronounced influence from meltwater (> 1.5% sea ice melt; Table 2) in the upper
30 m, with median dCo concentrations equal to 357.5 pmol L$^{-1}$ in the MIZ, though with large
variability (range 25.9-546.2 pmol L$^{-1}$) likely due to surface drawdown and additional dCo
sources. Surface concentrations in this region ranged from approximately 100-500 pmol L$^{-1}$ (Fig.
4D-F, M-N). The dCo in surface waters decreased slightly in the Makarov Basin and reached
some of the lowest observed concentrations at the North Pole (210 pmol L$^{-1}$; Fig. 4H; Fig. 5),
though concentrations were still slightly higher than at Station 1, the only Pacific station (Fig.
4A). Although some elements such as dFe showed noticeable elevated concentrations in the
vicinity of North Pole in surface waters compared to surrounding waters (L. Jensen, pers.
comm.), dCo remained lower than on the shelf and in the MIZ (Fig. 5). Surface dCo at the North
Pole was approximately 250 pmol L$^{-1}$, nearly half the concentrations observed in the Canada
Basin (Fig. 4H).

*3.2.3 Dissolved cobalt in Pacific halocline and deep waters*

While silicate (SiO$_3$) and phosphate (PO$_4^{3-}$) concentrations were indicative of the advection of
PHW (Fig. 2E, F), dCo did not show a prominent enhancement within this feature (Fig. 5A),
likely due to the slightly lower relative concentrations of dCo in Pacific waters compared to shelf
waters (station 1; Fig. 4A). Median concentrations of dCo in waters dominated by Pacific water
(> 95%) were 269.6 pmol L$^{-1}$ (range 64.1-687.3 pmol L$^{-1}$) while on the shelf they were 526.0
pmol L$^{-1}$ (Table 2). Any elevated dCo concentrations observed within the PHW density layer
($\sigma_\theta$= 26.2-27.2; Steele et al., 2004) was likely added along the flow path of Pacific water across
the Bering Shelf (Fig. 4B). Thus, stronger relationships were observed with other elements which
are also elevated on the shelf (e.g. dFe and dMn; M. Hatta pers. comm.) than with SiO$_3$ or other
macronutrients (e.g. PO$_4^{3-}$).

The dCo was remarkably constant within the deep Arctic, reflective of both AW and deep Arctic
bottom water (Fig. 5A; Swift et al., 1983). Concentrations in AW (> 95% AW, and all depths >
500 m) had a median value of 61.6 pmol L$^{-1}$ (Table 2), in between the average deep water dCo
concentrations found in the Pacific and Atlantic (Fig. 3). The near-bottom sample from some
profiles also showed slightly lower dCo (< 5 pmol L$^{-1}$) than the sample immediately above it
(Fig. 4C, D, F), perhaps indicating some influence of the weak nepheloid layers on bottom-water
scavenging of dCo in the Arctic (Noble et al., 2016).

*3.3 Labile cobalt distributions*



*3.3.1 Labile cobalt in surface waters*

LCo is the fraction of total dCo that is either not organically complexed or weakly bound by organic ligands, and represents the "labile" fraction of the total dCo pool either in terms of biological uptake or scavenging (Saito et al., 2004; Saito and Moffett, 2001). LCo distributions looked remarkably similar to dCo in the upper water column (Fig. 4, 5). Concentrations were lower than dCo, ranging from 0 (not detectable) to 600 pmol $L^{-1}$ on the Canadian side of the Chukchi Shelf (station 61, 66). LCo comprised 20-35% of the total dCo pool in the upper water column (Fig. 6), with the highest percentage of LCo found over the Chukchi shelf and approximately 20% LCo in Pacific waters (station 1; Fig. 6). LCo decreased more rapidly with respect to distance from the shelf than dCo in the Canada Basin and towards the North Pole, with the North Pole region containing significantly lower median concentrations of LCo (10.3 pmol $L^{-1}$, $p < 0.05$) than surrounding waters (148.0 and 117.0 pmol $L^{-1}$ on the shelf and MIZ, respectively; Table 2). The majority of the LCo appeared to either be removed via scavenging or biological uptake in the upper water column in the Canada Basin and along the Lomonosov Ridge. Some of the highest median LCo concentrations were observed in the upper 30 m in the MIZ and in waters containing significant sea ice melt (> 1.5%, Table 2), with median concentrations rivaling those on the shelf (Table 2). The LCo in these samples had a large range in many cases (48.8-233.0 pmol $L^{-1}$ in samples with > 1.5% sea ice melt), suggesting that sea ice may be a source of LCo, and that it is taken up quickly in surface waters after input from meltwater.

*3.3.2 Labile cobalt in Pacific halocline waters and deep waters*

LCo was extremely low, and often undetectable, in the deep waters of the Arctic (Fig. 4). Any detectable LCo at these depths represented less than 10% of total dCo (Fig. 6), with the majority of the dCo in the deep Arctic was strongly organically complexed. Similar to dCo, there was no observable enhancement of LCo in PHW, with LCo closely following that of dCo and other shelf-enhanced trace metals such as dFe and dMn (L. Jensen, pers. comm.). LCo decreased below the upper 250 m, and the median concentration of LCo in the Atlantic layer was 2.2 pmol $L^{-1}$ (Table 2) virtually equal to the detection limit of the method (2.1 pmol $L^{-1}$), suggesting scavenging or uptake of LCo in the upper water column and little to no detectable LCo in deep waters of the Arctic.

*3.4 Dissolved and particulate manganese and particulate cobalt distributions*

DCo and dMn had very similar distributions across the transect (Hatta et al. in prep). The pCo and pMn concentrations were slightly decoupled from the dissolved concentrations, with a subsurface peak in both (Fig. 7), as opposed to the surface peak observed in dCo and dMn (Hatta et al. in prep). The maximum in pCo and pMn occurred at depths of approximately 200-300 m, corresponding to a region of significantly elevated concentrations of particulate Mn-oxides (P. Lam pers. comm.). Overall, pCo and pMn concentrations were the highest on the shelf, with visible increases at the base of the profiles near the sediment water interface (Fig. 7B, C). Concentrations of pCo and pMn declined by almost an order of magnitude from the shelves into the Arctic basin, with concentrations ranging from 20-40 pmol $L^{-1}$ and 1-10 nmol $L^{-1}$ for pCo and pMn, respectively. Deep water (> 1000 m) particulate concentrations for both metals were





remarkably consistent, with concentrations varying slightly over the entire Arctic basin (Fig. 7D,
H). These deep water pMn and pCo concentrations are notably higher than in other regions, such
as deep Pacific waters (Lee et al., 2018).
*3.5 Modeling sensitivity experiments*
Overall, the control model agrees well with the data over a number of different depth strata (Fig.
8). In the surface layer (0-50m), the model output is most consistent with the observations (Fig.
8A), although in general, the model tends to produce maximum levels of dCo that underestimate
the highest dCo concentrations observed. Part of this is likely due to the fact that we are
comparing annual mean model output against synoptic scale in situ observations. However, the
model may also be underestimating sources of dCo in the Arctic. Below 50 m, there is also good
agreement with observations (Fig. 8B), with the model capturing the much lower dCo
characteristic of these waters and in particular the contrast between our data in the Arctic and
other data from the North Atlantic (Dulaquais et al., 2014). In the deepest layers (Fig. 8C and D),
the model again is able to reproduce the decline in dCo to ~60 pmol L$^{-1}$ and the consistency
between the deep Arctic and North Atlantic.
In order to capture the major processes contributing to the modeled dCo sources and sinks, the
proportion of the dCo signal in two distinct depth horizons was further explored using a set of
sensitivity experiments. In the 0-50 m depth range (Fig. 9), rivers in the model have no large
scale impact on the Arctic-wide dCo signal (Fig. 9A), while removing sediment margin sources
reduced dCo by over 80%, (Fig. 9B). The strong effect of sediment Co supply in the model is
largely driven by the large shelf area in the Arctic. In contrast to the eastern tropical Pacific
oxygen minimum zone where low oxygen concentrations contributed significantly to the source
of dCo (Tagliabue et al., 2018) due to reductive dissolution of sedimentary Mn-oxides (Hawco et
al., 2016), enhanced sediment Co supply under low oxygen had no impact in the Arctic, due to
higher levels of oxygen typical of this basin. Similarly, modulating the effect of oxygen on Co
scavenging also had little impact in the Arctic (Fig. 9C). It was notable that keeping bacteria
scavenging constant (e.g. eliminating the effect of changes in bacterial biomass on scavenging)
reduced dCo at the surface by over 60% in some places, indicating that lower rates of scavenging
was also contributing to the high rates of dCo in the surface ocean (Fig. 9D). Thus, our model
experiments suggest that the high levels of dCo in the Arctic surface waters are due to
sedimentary supply, with a secondary role played by reduced scavenging due to low rates of
activity associated with Mn-oxidizing bacteria due to colder temperatures. In the 700-800 m
depth horizon, we similarly find that changing sediment supply is more important than rivers
(Fig. 10A and B), but that the effect of sediments is reduced compared to the surface. Equally,
retardation of Co scavenging under low oxygen has a minor role in the ocean interior (Fig. 10C),
with bacterial biomass again having a significant effect on the interior dCo signal (Fig. 10D).
Thus in contrast with the surface, we find that in the 700-800 m stratum there is a roughly equal
role played by sediment Co supply and low rates of Co removal by Mn-oxidizing bacteria in
maintaining the dCo signal. These results are different to similar assessments in the southern
equatorial Pacific where the lower oxygen levels typical of this region led to an enhanced role for
sediment supply linked to low oxygen and reduced Co scavenging under low water column
oxygen in driving high levels of dCo (Tagliabue et al., 2018).





**4. Discussion**
*4.1 Quantifying external sources of cobalt to the Arctic Ocean*
The coherence of the dCo and LCo distributions with that of dMn, along with evidence from the
model output, suggest that the extensive shelf sediments in the Arctic are the dominant source of
Co in the Canadian section of the Arctic Ocean (Fig. 5, 9). Mn is known to be an excellent tracer
of sediment input due to the high solubility of reduced Mn emanating from reducing sediments
(Johnson et al., 1992; März et al., 2011; McManus et al., 2012; Noble et al., 2012). By using the
dMn concentrations as a tracer for shelf input, we can quantify the proportion of the variance in
the dCo and LCo observations that are explained by this proxy for shelf input. Linear regressions
between dCo or LCo distributions and dMn in the upper 200 m across all of the stations explains
73% and 79% of the variance in the dCo and LCo concentrations, respectively (Fig. 11A; $p <$
0.05). This trend is driven primarily by the data in the upper 50 m. The variance explained
decreases however, if only the shelf stations (stations 2-10, 57-66) are included in the analysis
(data not shown), suggesting that some process other than shelf inputs couples the dMn and Co
distributions within the basin. The amount of the variance in the Co distributions that is
explained by shelf inputs is slightly less than that observed in the model (Fig. 9B), though both
agree that shelf inputs are the dominant source.
The modeling results suggest that nearly all of the dCo in the upper 50 m can be accounted for
by a combination of a sediment source and diminished scavenging (Fig. 9B and D). However,
the inferences from observations suggest that 20-30% of the variance cannot be explained by a
shelf source alone. If the dCo and LCo is examined against salinity for all stations in the transect
in the upper 200 m, then salinity can explain 24% and 28% of the variance for dCo and LCo,
respectively (data not shown). This relationship is improved if only the stations in the central
Arctic basin are included (stations 30-43), and then salinity explains 47% of the dCo and 57% of
the LCo distributions (Fig. 11B). The coherence of dCo and LCo with salinity across the dataset,
and particularly in this region, appears to be due to a contribution of low salinity water from
rivers, rather than from sea ice melt (Fig. 12C), as no relationship was observed with the fraction
of sea ice melt determined from $\delta^{18}$O isotopic measurements of seawater (Bauch et al., 2005;
Cooper et al., 2005, 1997). Instead, the relationship with salinity is driven by freshwater inputs
from rivers, as a strong relationship is observed with the fraction of meteoric water (Fig. 12D).
These stations correspond to a region of anomalously high dFe and DOC concentrations (Jensen
et al., in prep, D. Hansell, pers. comm.), interpreted to be indicative of river inputs carried across
the basin in the Transpolar Drift (TPD) (Gascard et al., 2008; Klunder et al., 2012; Wheeler et
al., 1997). This is supported by measurements of $^{228}$Ra, which track shelf inputs throughout the
Arctic due to interactions between the sediment-water exchange processes (Kipp et al., 2018; van
der Loeff et al., 2018). A similar relationship was also observed with salinity in the North
Atlantic, supporting the role of rivers as a source of dCo (Saito and Moffett, 2001). In our model
sensitivity experiments, we found a small effect of rivers on dCo (Fig. 9A and 10A), and the
Co/N river endmember in the model was similar to that measured by the Arctic Great Rivers
Observatory (Holmes et al., 2018). It appears that the data suggests a larger role for rivers than
what is captured by the model, which could imply that gross riverine fluxes are underestimated
by our model. However it is difficult to disentangle riverine processes from other processes
happening on the shelf like groundwater inputs (Charette et al., in review). It is possible that



there is some mixing of river and sediment dCo occurring in the coastal zone or that our global
scale model is not able to properly account for the physical transport of fluvial signals into the
open basin.
The presence of such high concentrations of trace elements and isotopes at the North Pole was
surprising, yet several tracers indicate that this is an area significantly influenced by river and
shelf input from the surrounding continents (Kipp et al., 2018; van der Loeff et al., 2018). The
elevated concentrations of dCo at great distances from the continental shelf is also likely partially
due to the enhanced organic complexation of dCo in TPD waters. Averaged over the entire
dataset dCo is 79±13% organically complexed (21±13% labile) in the upper 200 m of the water
column. However, at TPD influenced stations (stations 29-34; Charette et al. in review), dCo is
92±6% organically complexed, significantly higher than in the rest of the transect (*paired*
*sample t*-test, $p < 0.05$). This suggests that elevated concentrations of DOC from Arctic rivers
entrained in the TPD or ligands produced in-situ may play a role in stabilizing a portion of the
dCo pool during transport towards the North Pole. Although the exact character of the organic
dCo-binding ligands in seawater are unknown, in the Arctic it is likely that humic-like
substances contribute some portion of the organic complexation observed, due to the presence of
elevated colored dissolved organic matter (CDOM) in the TPD (Wheeler et al., 1997), consistent
with the presence of humic substances (Del Vecchio and Blough, 2004). Despite the presence of
humic substances, it seems somewhat unlikely that humics account for all of the ligands
complexing dCo in this region. Our analytical method distinguishes organically-bound Co as the
fraction of total dCo that is more strongly complexed than our competing ligand (DMG). The
complexation of humic and fulvic-like substances with Co has been shown to be much weaker
than the Co(DMG)$_2$ complex ($\log K^{cond}_{Co(HS)} \sim 8$ versus $\log K^{cond}_{Co(DMG)_2} = 11.5 \pm 0.3$; Yang and Van
Den Berg, 2009). Ligands similar to those suspected to complex Co in open ocean waters of the
Atlantic or Pacific could be responsible for Co stabilization in the TPD waters (Saito and
Moffett, 2001). These ligands are presumed to have functional groups similar to cobalamin
(vitamin B$_{12}$), with a Co atom tightly bound inside a corrin ring. Cyanobacteria and some
archaea are known cobalamin producers (Bertrand et al., 2007; Doxey et al., 2015; Heal, 2018;
Heal et al., 2017; Lionheart, 2017) and both are found in the Arctic (archaea; Cottrell and
Kirchman, 2009; cyanobacteria; Waleron et al., 2007; Zakhia et al., 2008), although in very low
abundance. The nature of the organic molecules binding dCo in this region will be interesting to
explore further in future studies.
Overall, both the modeling results and observations agree that the dominant source of Co in the
Arctic is from the extensive shelf sediments surrounding the Arctic Ocean, with additional
contributions from Arctic rivers. The observations however, show that sources vary in
importance in space, with sediment sources clearly dominating in stations close to the shelf, and
river sources dominating in the central Arctic basin through the influence of the TPD. Whether
or not shelf sediments act as a capacitor, storing and then releasing Co to overlying water, for
terrestrially derived riverine sources or for Co delivered to sediments in sinking marine organic
matter remains unknown. It is clear however, that the riverine source dominates the distribution
observed near the North Pole where dCo and LCo concentrations remain high despite the
distance from land, and that organic complexation likely plays a role in the distal transport of this
dCo (Charette et al., in review).





*4.2 Cobalt scavenging and internal cycling*

A striking feature of the dCo and LCo dataset is the vertical transition in the water column from high to low Co concentrations throughout the deep Arctic (Fig. 5). The question remains whether or not 1) this elevated dCo is scavenged at a shallow depth horizon, or 2) if the high dCo concentrations in the surface layer (< 200 m) are simply physically isolated from deeper water masses, or a combination of the two. This would suggest that the Atlantic water characteristic of the deep Arctic doesn't mix with the modified surface Arctic water containing high concentrations of Co. We examined both hypotheses within a modeling framework and compared this to the observations. In the model, the dCo is scavenged primarily in the upper 50 m with almost no scavenging below 200 m (data not shown). The dCo scavenging in the model is primarily controlled by Mn-oxidizing bacteria, which have a strong temperature dependence in the model (Tebo et al., 2004). The cold temperatures in the majority of the Arctic prevent enhanced scavenging of dCo by this mechanism in the Arctic compared to other basins (Hawco et al., 2018; Saito et al., 2017; Tagliabue et al., 2018). However, relatively warmer temperatures on the shallow shelves suggest that scavenging is enhanced in this region (Fig. 4), and the coherence of the pCo and pMn peaks in the upper 200-250 m (Fig. 7) support this mechanism of upper ocean scavenging. Evidence from [234]Th data shows very little (to no) particulate organic carbon (POC) flux in the upper water column along this transect, however strong lateral transport from the shelves to the basin was observed (Black et al. 2018). This lateral transport was observed both in the upper water column and at depth, suggesting fast-moving currents through the deep canyons may be significant in transporting material from the shelf into the basin (Black et al. 2018). It is possible that additional scavenging of Co may occur along this flow path. Some of the profiles observed in the deep basin also show evidence for deeper ocean scavenging in the Atlantic water (e.g. Fig. 4E, H, P).

Additional insights on Co scavenging in this basin can be observed by exploring the dCo: phosphate (P) ratios (pmol L$^{-1}$:μmol L$^{-1}$) along the transect (Fig. 13). The relationship between dCo and P in the Arctic water column yields insights into biological uptake and regeneration processes acting on the dCo inventory, as well as scavenging. An analysis completed by Saito et al. (2017) showed that positive slopes in the dCo:P relationship were indicative of regeneration, while negative slopes were indicative of biological uptake or scavenging (Saito et al., 2017). The high dCo in the Arctic yields a unique dCo:P relationship compared to the North Atlantic (Fig. 13A; Saito et al., 2017). When dCo:P slopes ($r^2 > 0.6$) are binned according whether they are positive (Fig. 13B) or negative (Fig. 13C) and then plotted with depth (Fig. 13D), a few patterns are apparent. Positive dCo:P slopes are observed largely within a confined depth strata in the PHW (Fig. 13D). This is not surprising, given that deep Pacific waters carry a strong regeneration signal. However, at most other depths in the water column the dCo:P slopes are negative, showing that scavenging is occurring to some extent throughout the water column (Fig. 13D). With one exception, the magnitude of the negative dCo:P slopes are greater in the upper water column, supporting the model results and our interpretations of the pCo profiles that most of the scavenging occurs in the upper water column but also continues to occur throughout the deep Arctic.

This evidence, combined with the coinciding maxima observed in pCo and pMn, suggest that a significant amount of scavenging occurs in the upper water column, but that additional



scavenging continues to occur below these depths. The elevated pCo concentrations in the deep
Arctic compared to other regions (Lee et al., 2018) suggest that scavenging over long timescales
continue to add to the pCo pool. This mechanism likely prevents high concentrations of dCo to
penetrate below the shallow modified Arctic surface water. However, it is clear that there is very
little mixing between the modified surface waters, the PHW, and the deep Atlantic water in the
Arctic (Steele et al., 2004). Thus, it is likely a combination of upper ocean scavenging, and little
mixing between water masses, that keeps the elevated dCo and LCo confined to the surface
waters in Arctic, yielding the intense scavenged-like profile of Co in this region compared to
other basins (Fig. 3).
*4.3 Increases in Co inventories over time in the Canadian sector of the Arctic Ocean*
Samples collected on the shelf in the Beaufort Sea in 2009 in proximity to the U.S.
GEOTRACES transect in 2015 (Fig. 1) had significantly lower dCo (*paired t*-test, $p < 0.05$) than
shelf samples from 2015 (Fig. 14). Shelf samples for dCo from 2015 were approximately 3.5
times higher than the dCo and approximately eight times higher in LCo than in 2009 (Fig. 14C).
The maximum dCo concentration measured in 2009 was 301 pmol L$^{-1}$, while in 2015 it was 1852
pmol L$^{-1}$. The dCo and LCo concentrations below 150 m agreed very well however, between the
two years (Fig. 14A, B). Several factors could account for the higher dCo and LCo observed in
2015 compared to 2009. The Co samples from 2009 were unfiltered, and were not stored with
gas-absorbing satchels like the samples from 2015. Recently, loss of dCo has been observed in
the presence of oxygen during storage, however this loss was most pronounced for samples in
low oxygen regions (Noble, 2012). The mechanism of the dCo loss is unknown and is difficult to
quantify from these samples, however the waters are well oxygenated in this region (Fig. 2B)
and thus the loss due to storage was likely minimal. However, we can not say for certain how
much of the observed increase in dCo over time is due to a storage artifact. Previous work has
shown a maximum loss of dCo of 40% after 5 months of storage (Noble, 2012). If we consider
that 40% of the dCo could have been lost in the samples collected from 2009, the data from 2015
still show an increase in dCo of approximately 400%. Thus, although we can not quantify with
certainty the percent increase in dCo over time in the Canadian sector of the Arctic, it is likely
that it is still significant.
The increase in dCo over time in the Arctic is interesting, and has been documented for other
tracers in the Arctic. Kipp et al. (2018) and van der Loeff et al. (2018) noted that $^{228}$Ra has
increased over time in the central Arctic. They suggest that increases in shelf and/or river inputs
from thawing permafrost are the source of this elevated $^{228}$Ra (Kipp et al., 2018; van der Loeff et
al., 2018). A similar mechanism is likely increasing metal inventories over time on Arctic
shelves. The majority of the variance (~70%) in dCo in the upper 100 m on the U.S.
GEOTRACES transect could be explained by a shelf source, and the remainder was likely
associated with river inputs (Fig. 11). If these sources are similar to the sources of dCo in 2009,
then an increase in either a shelf or river flux could be responsible for the dramatic increase in
dCo over time. There is not enough data to state whether the river dCo flux has changed over
time in the Arctic, however several studies have documented an increase in river discharge over
time due to increases in permafrost melt (Doxaran et al., 2015; Drake et al., 2018; Kipp et al.,
2018; van der Loeff et al., 2018; Tank et al., 2016; Toohey et al., 2016). The increase in river
discharge has the potential to considerably increase trace metal inventories in the future Arctic





Ocean, perhaps particularly for those metals that are strongly organically complexed, thus
protecting against scavenging in the estuarine mixing zone (Bundy et al., 2015). These increases
in metals over time will have implications for metal stoichiometries and phytoplankton growth in
a changing Arctic Ocean.
*4.4 Implications of the Arctic as a net source of Co to the North Atlantic Ocean*
The concentrations of dCo and LCo in this region of the Arctic are some of the highest
concentrations that have been observed thus far in the ocean. In some cases, the dCo was almost
ten times higher than in the low oxygen region of the Eastern Pacific (Hawco et al., 2016).
Although the Arctic is considered to be a macronutrient poor system, in contrast to other
oligotrophic regions the Arctic is quite enriched in micronutrients (Jensen et al., 2019; Jensen et
al., in prep). These distinct nutrient ratios may have implications for Arctic phytoplankton
communities, as well as communities in the North Atlantic that are influenced by inputs from the
Arctic.
Arctic waters are thought to primarily exit the basin and impact the North Atlantic via the
Canadian archipelago and the Fram and Davis Straits (Talley, 2008). The organic complexation
and stabilization as well as the high concentrations of dCo suggest that some of this dCo might
exit the Arctic and impact nutrient distributions in the North Atlantic. Noble et al. (2016) noted a
plume of elevated dCo in the western portion of the U.S. GEOTRACES North Atlantic (GA03)
transect that did not correspond with a signature from reducing sediments as on the North
Atlantic eastern boundary. Noble et al. (2016) postulated that high dCo in Labrador Seawater
(LSW) was the source of this signal, due to the presence of a corresponding signature of low
silica that is characteristic of this water mass. The authors noted that the anomalously high dCo
could be from elevated dCo in Arctic waters, or due to high dCo on the shelf that is picked up
along the flow path of the LSW, or a combination of the two (Noble et al., 2016). Our data
suggests that likely a combination of the high dCo observed in this study and additional Co
entrained on the shelf in the Labrador Sea contribute to that signal, and when observed in
temperature and salinity space the data supports this hypothesis (Fig. 15). The Arctic source
waters that contribute to the formation of LSW have a low salinity signature, and are likely
significantly modified as they exit the Canadian archipelago, Fram Strait and Davis Straits
(Yashayaev and Lodor, 2017). From this data we cannot quantitatively connect the elevated dCo
and LCo observed in the Arctic source waters to the LSW seen in the western Atlantic, given the
complex history (e.g. transformation, mixing) of source waters in the Labrador Sea region (Le
Bras et al., 2017). However it is apparent that the low salinity Arctic waters contain high Co
(Fig. 15), which given the advective pathways of these water masses from the Arctic, suggests
that they may act as a source of Co to lower latitude waters. Interestingly, the high dCo in the
Arctic has a distinct LCo/dCo signature compared to that observed in the western North Atlantic
(Fig. 15A). Due to the significant impact that Arctic shelves and rivers have on the dCo signature
observed in this study, it is likely that additional Co may be added to these waters as they pass
through the Canadian archipelago. Additional LCo is likely entrained in this water mass as the
surface Arctic water moves through the Canadian archipelago. The fate of these waters and their
Co as they exit via the Fram and Davis Straits is unknown. Constraining these Arctic
endmembers and how they contribute to dCo distributions in the North Atlantic deserves further



attention, as it has interesting implications for nutrient resource ratios for North Atlantic
phytoplankton communities.
The possibility that elevated micronutrient concentrations from the Arctic are being exported to
the North Atlantic could have interesting implications for phytoplankton nutrient utilization and
community composition. DCo and dZn for example, which can be interchanged within carbonic
anhydrase in some eukaryotes (Lane and Morel, 2000; Sunda and Huntsman, 1995; Yee and
Morel, 1996), are elevated in the Arctic compared to the North Atlantic and South Pacific (Fig.
16A, B). The higher concentrations of both metals results in a dCo/dZn ratio that is quite similar
to that observed in the North Atlantic, however the range in this ratio is large (Fig. 16C). Small
changes in the sources of each of these metals could manifest as big impacts on the ratio of these
micronutrients in surface waters. However, the shift in these micronutrient ratios may not impact
the resulting phytoplankton quotas (Figure 16D). Despite quite different total concentrations of
dCo and dZn in the Arctic compared to the North Atlantic for example, the measured quotas are
quite similar (Twining et al. in prep, Figure 16D). However, if river inputs continue to increase
with an increase in permafrost thawing in the warming Arctic (Jorgenson et al., 2006) and
similar increases in dCo are observed over time as seen in this work, then total metal inventories
in the Arctic may begin to influence the North Atlantic to a greater extent. These source changes
may disproportionately affect Co compared to Zn, whose primary source was found to be from a
regeneration signal on the shelf rather than from river input (Jensen et al., 2019), and whose total
inventory is small compared to Zn. Understanding how future changes in metal sources in the
Arctic may impact the North Atlantic or shifts in phytoplankton community structure will be
important to constrain.
**5. Conclusions**
The unique dissolved and labile Co distributions observed in the Arctic have noteworthy
implications for Arctic ecosystems and for future changes in micronutrients in the warming
Arctic. Sediment and river inputs to the Arctic appear to be the dominant mechanisms for the
input of dCo to the Arctic, and these elevated signals persist over a broad area of the western
Arctic far from their source regions. This appears, at least in part, to be due to the relatively slow
scavenging of Co in this basin that is suggested from modeling outputs to be related to the low
temperatures and slower kinetics of Mn-oxide formation. The majority of this scavenging
appears to happen on the shelf with an advected signal of particulate Mn and Co appearing into
the Arctic basin. Co was also shown in this work to be increasing over time on the shelf in the
Canadian Arctic, possibly due to increases in river inputs from thawing permafrost, though this is
difficult to constrain in the present dataset. Given the significant increase in Co over time in the
Arctic and the modification of low-salinity Arctic waters as they exit the Arctic into the North
Atlantic and the Labrador Sea, it is difficult to determine if there is a net flux of Co out of the
Arctic and into the North Atlantic, however evidence in this work suggests that the distinct Co
waters of the Arctic likely impact downstream micronutrient concentrations. These impacts are
likely to become increasingly important in the future, with increased warming and changes to Co
sources in the Arctic basin.
**6. Author contributions**





RMB analyzed the samples and wrote the manuscript. MRC developed the data processing code and helped write the manuscript. MAS designed the study and helped write the manuscript. AT, NJH, PLM, BST, MH, AN, SGJ, and JTC contributed data and helped write the manuscript.

**7. Data availability**

The data for this manuscript are available through BCO-DMO for GN01 (https://www.bco-dmo.org/project/638812) and through BODC for GIPY14 (https://www.bodc.ac.uk/geotraces/data/inventories/0903/). The dissolved and labile cobalt data for GN01 specifically is available at https://www.bco-dmo.org/dataset/722472.

**8. Acknowledgements**

We would like to thank the captain and crew of the USGC *Healy*, Gabi Weiss and Simone Moos for sampling, and Dawn Moran, Noelle Held and Matt McIlvin for help with sample preparations and analyses, Dr. Ana Aguilar-Islas and Dr. Robert Rember for small boat and sea-ice hole operations, the Ocean Data Facility at Scripps Institution of Oceanography for macronutrient, oxygen, and salinity measurements, S. Rauschenberg for sample collection, and P. Schlosser, R. Newton, T. Koffman, and A. Pasqualini for water mass fraction data. This work was supported by NSF-OCE #1435056 to M. Saito and S. John, as well as Woods Hole Oceanographic Institution Postdoctoral Scholar grant to R.M. Bundy and M.R. Cape. M. Hatta was supported by NSF-OCE #1439253. A. Tagliabue was supported by the European Research Council (ERC) under the European Union's Horizon 2020 research and innovation programme (Grant agreement No. 724289). PM and BT were supported by NSF-OCE #1435862 and PM was also supported by the National High Magnetic Field Laboratory (PLM). PLM is supported by the National Science Foundation through DMR-1644779 and the State of Florida. J.T. Cullen was supported by the Natural Sciences and Engineering Research Council (NSERC) of Canada and an International Polar Year (IPY) Canada grant.





**Table 1:** Average dCo concentrations from blank, internal standard, and consensus standard runs.

|  | n | dCo (pmol L$^{-1}$) | std dev |
|---|---|---|---|
| blank | 29 | 2.5 | 0.7 |
| internal standard | 26 | 50.3 | 7.6 |
| SAFe D1 | 3 | 47.9 | 2.1 |
| SAFe D2 | 3 | 45.2 | 2.1 |
| GSP | 3 | 2.4 | 1.8 |
| GSC | 3 | 77.9 | 2.8 |





**Table 2:** Median, maximum and minimum concentrations of total dissolved (dCo) and labile cobalt (LCo) in samples with representative water masses and sources in the Arctic Ocean. Median concentrations were determined in each water mass type by using water masses that contained > 95% Atlantic water, > 95% Pacific water, > 10% meteoric water, and > 1.5% sea ice melt. Shelf stations were stations 2-10 and 60-66, MIZ stations 10-17 and 51-57 (< 30 m), and North Pole stations 27-36 (< 200 m). Ice hole samples were sampled from 1 and 5 m. The notation 'nd' means not determined.

| | dCo (pmol L$^{-1}$) | max | min | $n$ | LCo (pmol L$^{-1}$) | max | min | $n$ |
|---|---|---|---|---|---|---|---|---|
| Atlantic | 61.6 | 126.3 | 36.9 | 37 | 2.2 | 5.8 | 0.2 | 27 |
| Pacific | 269.6 | 687.3 | 64.1 | 41 | 45.8 | 133.8 | 2.5 | 35 |
| Meteoric | 266.1 | 497.2 | 64.1 | 27 | 77.5 | 139.8 | 11.6 | 25 |
| Shelf | 526.0 | 1852.1 | 25.9 | 30 | 148.0 | 578.7 | 6.1 | 30 |
| MIZ | 357.5 | 546.2 | 25.9 | 19 | 117.0 | 158.6 | 6.1 | 19 |
| North Pole | 139.8 | 280.2 | 64.2 | 14 | 10.3 | 22.0 | 1.5 | 14 |
| sea ice melt | 526.0 | 1021.5 | 207.3 | 3 | 151.1 | 233.0 | 48.8 | 3 |
| ice hole | 281.1 | 316.2 | 259.4 | 4 | nd | nd | nd | 4 |



**Figure Captions**

**Figure 1**: Standard CTD sampling stations (green) and trace metal rosette (TM) sampling stations (blue) from the GN-01 expedition in 2015, and trace metal sampling locations from the GIPY14 expedition in 2009 (red).

**Figure 2**: *In-situ* temperature (A), nitrate (B), salinity (C), phosphate (D), oxygen (E), and silicate (F) with neutral density anomaly contours from the northern and southern legs of the GN-01 transect as shown in Figure 1. Major water masses are labeled as modified Pacific Water (mPW), Pacific Halocline Water (PHW) and Atlantic Water (AW).

**Figure 3:** Median dCo concentrations at specific depth intervals from the Arctic Ocean (this study; red circles), Atlantic Ocean (blue triangles), and the Pacific Ocean (orange squares). Shaded regions indicate the upper and lower quartiles of the data in each dataset.

**Figure 4**: Dissolved cobalt (dCo; black circles) and labile cobalt (LCo; open circles) from all stations from the 2015 (A-Q) and 2009 (R) studies.

**Figure 5:** (A) dCo concentrations and (B) LCo concentrations in the Arctic Ocean.

**Figure 6**: The ratio of LCo (pmol L$^{-1}$) to total dCo (pmol L$^{-1}$) along the transect from south to north in the upper 1000 m.

**Figure 7:** Particulate manganese (pMn; open circles) and particulate cobalt (pCo; x) from several stations along the northern (A-D) and southern (E-H) legs of transect, with the same station designations as in Figure 4.

**Figure 8:** Model output (colors) compared to observations (dots) from 0-50 m (A), 50-150 m (B), 700-800 m (C) and 1500-2000 m (D).

**Figure 9:** (A) Model output of the proportion of the dCo signal from 0-50 m that is controlled by (A) rivers, (B) sediment input, (C) oxygen concentrations, and (D) removal by Mn-oxidation from Mn-oxidizing bacteria.

**Figure 10:** (A) Model output of the proportion of the dCo signal from 700-800 m that is controlled by (A) rivers, (B) sediment input, (C) oxygen concentrations, and (D) removal by Mn-oxidation from Mn-oxidizing bacteria.

**Figure 11:** dCo (closed circles) and LCo (open circles) in the upper 200 m plotted against (A) dMn in shelf stations only (stations 2-10, 57-66), as well as (B) salinity from only the stations influenced by the Transpolar Drift (stations 30-43).

**Figure 12:** dCo and LCo from select stations versus (A) the fraction of Atlantic water (F$_{atl}$; all stations < 500 m), (B) the fraction of Pacific water (F$_{pac}$; all stations < 500 m), (C) fraction of sea ice melt (F$_{ice}$; < 100 m and south of 84ºN) and (D) the fraction of meteoric water (F$_{met}$; < 500 m and north of 84ºN).

**Figure 13:** (A) The dCo (pmol L$^{-1}$) compared to phosphate (dP; μmol L$^{-1}$) from the GN01 dataset. (B) 5-point two-way linear regression of positive dCo:P slopes ($r^2 > 0.6$). (C) 5-point two-way linear regression of negative dCo:P slopes ($r^2 < -0.6$). (D) Depths where either a positive (blue) or negative (red) dCo:P slope was identified in the GN01 dataset. Additional details on the regression analysis can be found in Saito et al., (2017).

**Figure 14:** The dCo on the shelf measured in 2009 (GIPY14; black triangles) and 2015 (GN01; blue circles) in the upper 3500 m (A) and upper 500 m (B). Average and dCo and LCo in the upper 150 m from 2009 (grey) and 2015 (blue). Error bars represent the standard deviation and a (*) denotes a significant difference.

**Figure 15:** (A) The ratio of LCo to dCo (colors) from this study and the western portion of the GA03 North Atlantic transect (Noble et al., 2016) along with dCo concentrations (B) in temperature-salinity space, with Labrador Sea Water (LSW) source waters (solid black box) and





signature in the Atlantic (dashed box) are highlighted. (C) Sampling region in this study and the
stations used from Noble et al. (2016).
**Figure 16:** Median dCo concentrations (A), dissolved Zn concentrations (B) and dCo/dZn ratios
(C) in the upper 200 m in the Arctic (this study), North Atlantic (Noble et al., 2016), and in the
Southern Eastern Pacific (Hawco et al., 2016). (D) Co/Zn ratios in phytoplankton from the Arctic
and North Atlantic (Twining et al., in prep). Whiskers represent the lower (25%) and upper
(75%) quartiles.





**Figure 1.**

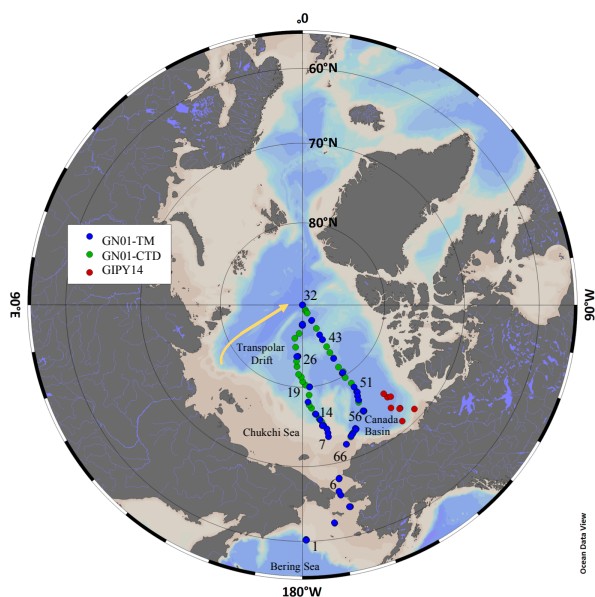





**Figure 2.**

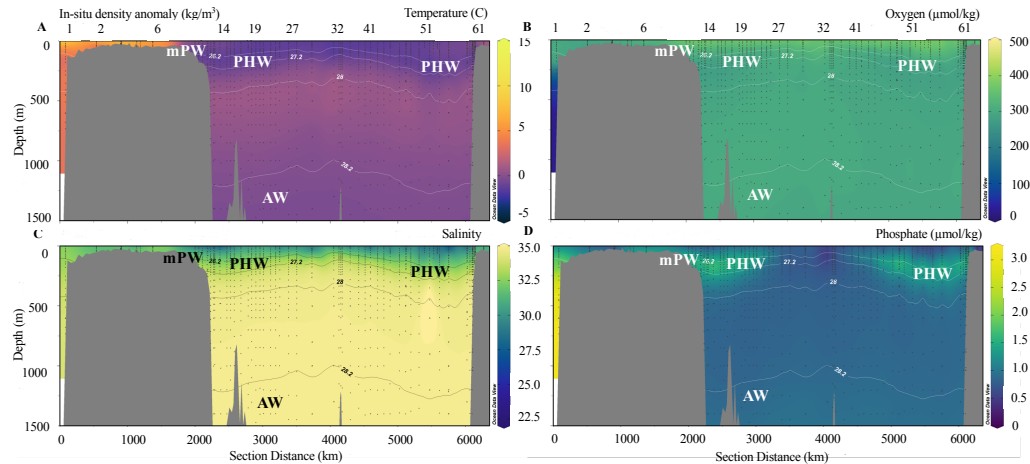





**Figure 3.**

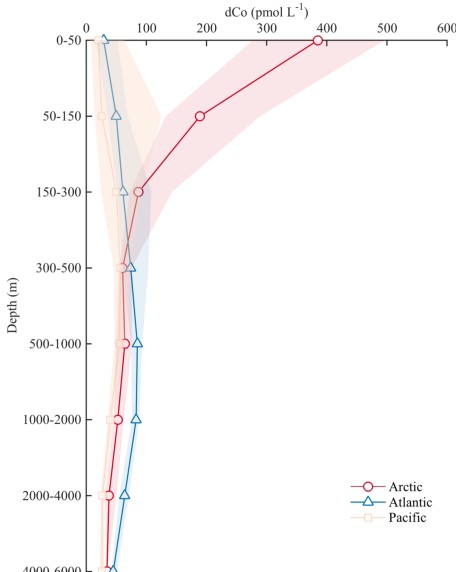



**Figure 4.**

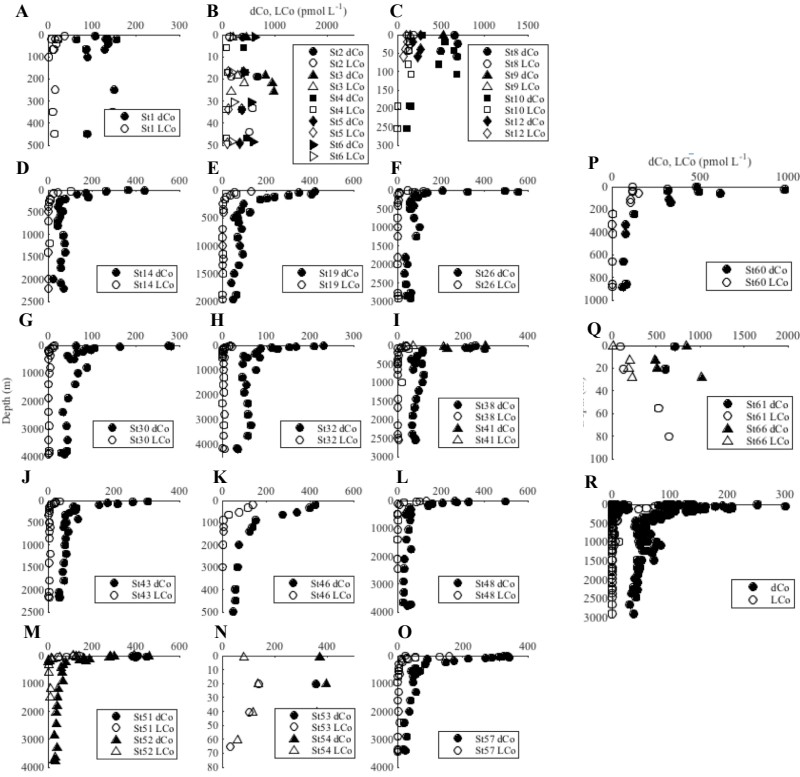



**Figure 5.**

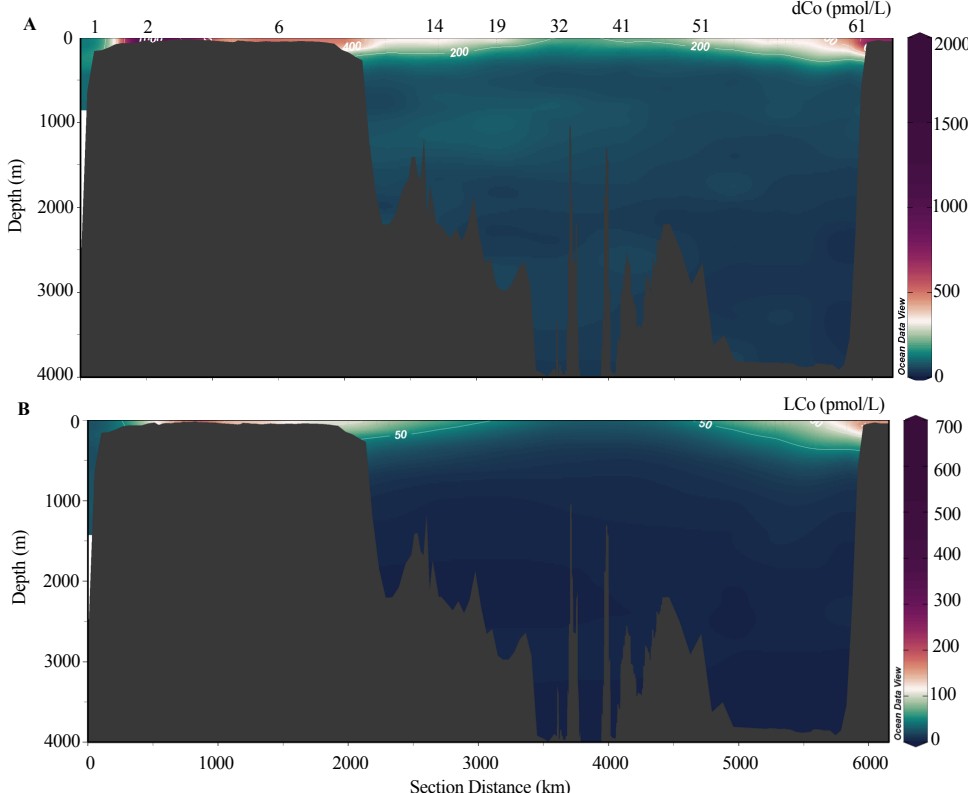




**Figure 6.**

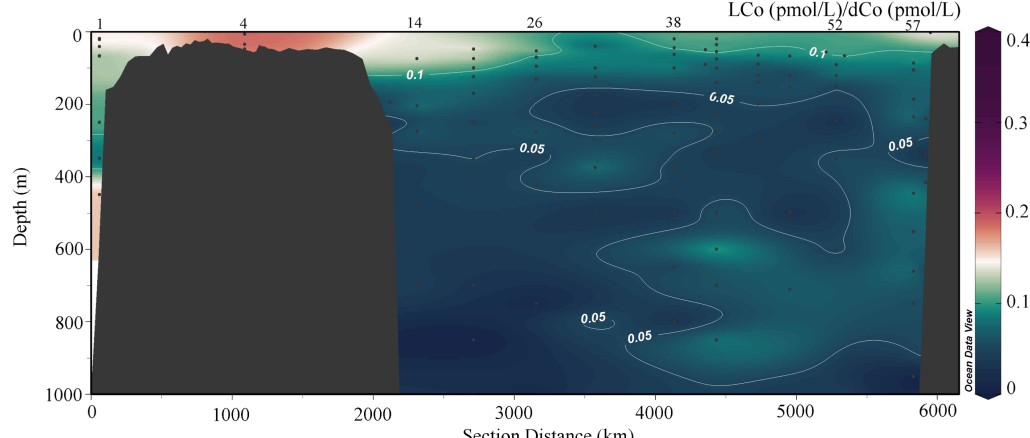



**Figure 7.**

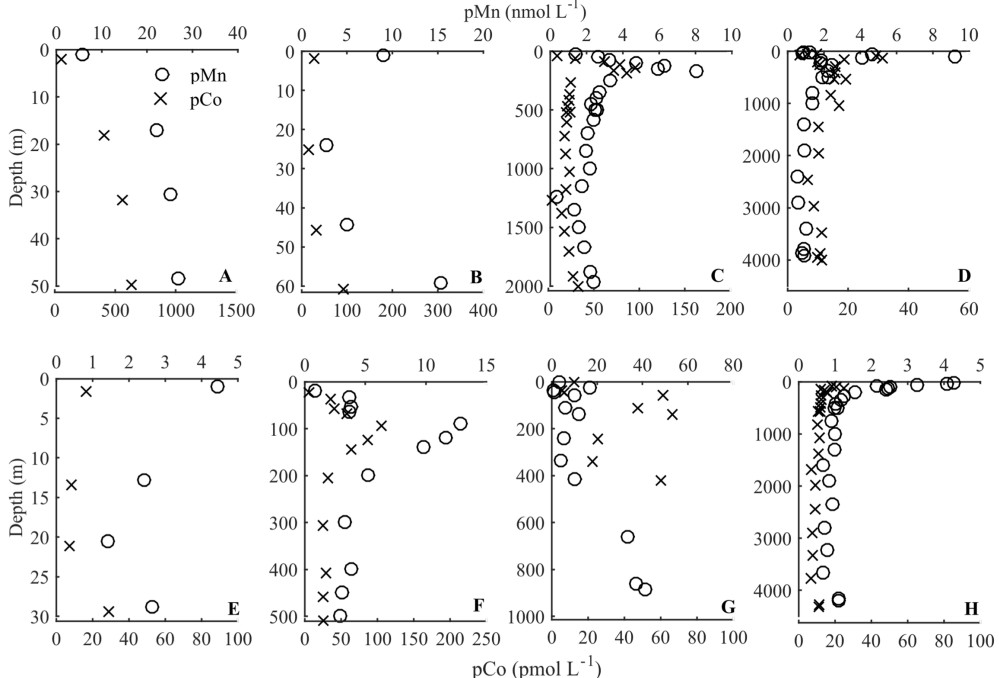





**Figure 8.**

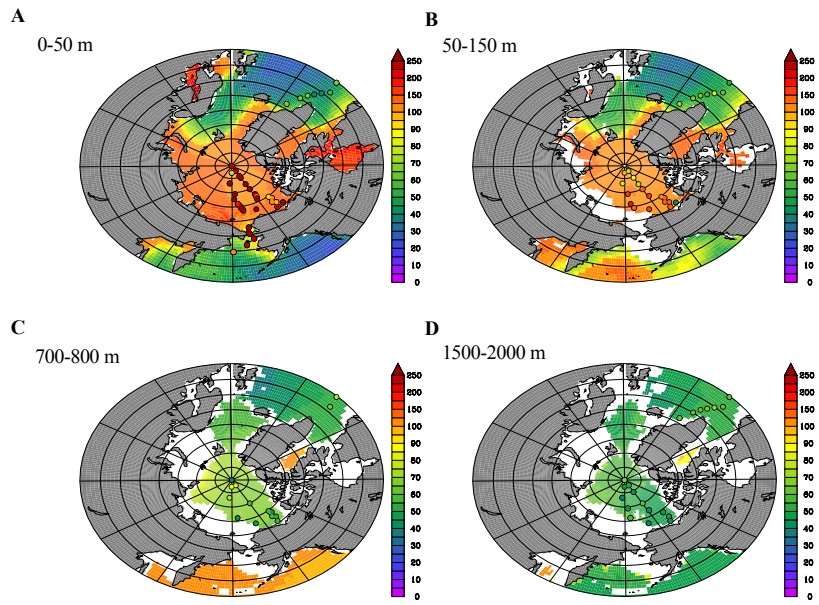





**Figure 9.**

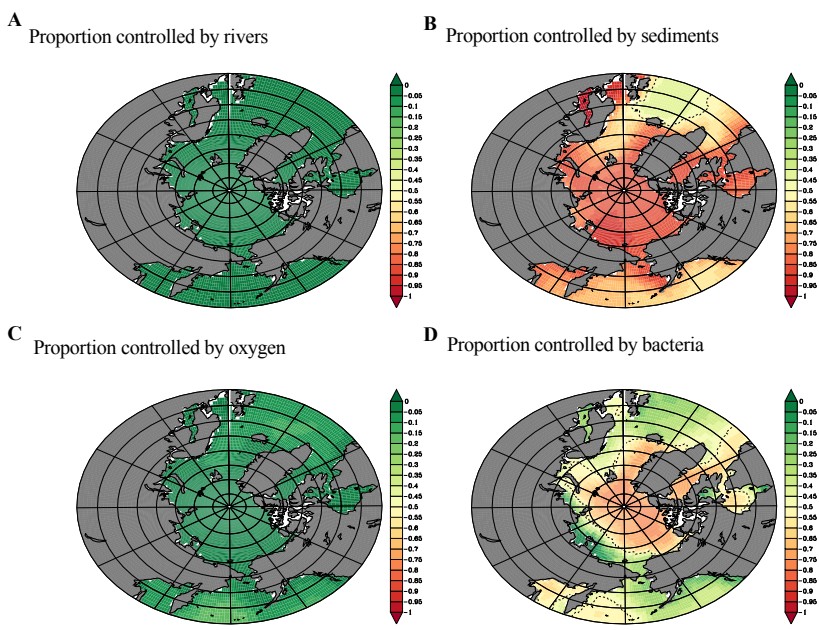




**Figure 10.**

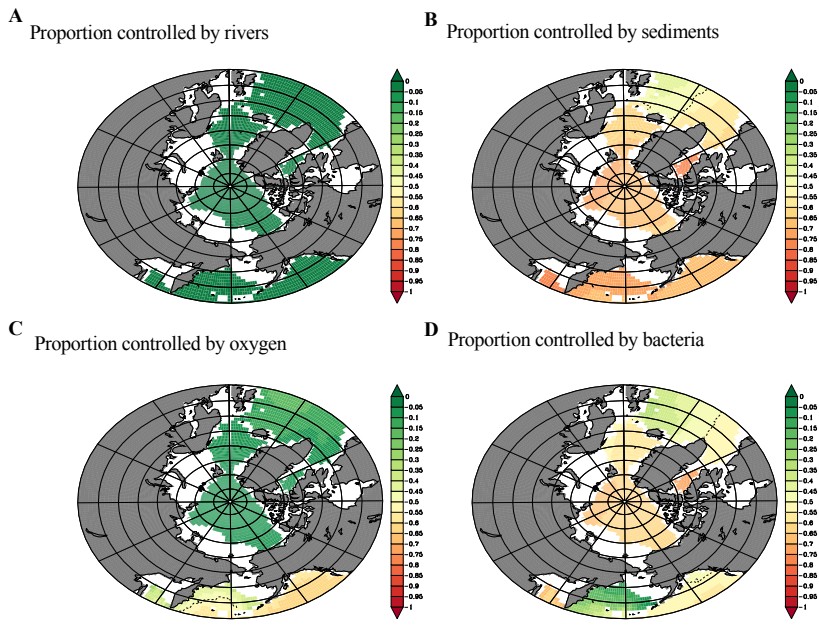





**Figure 11.**

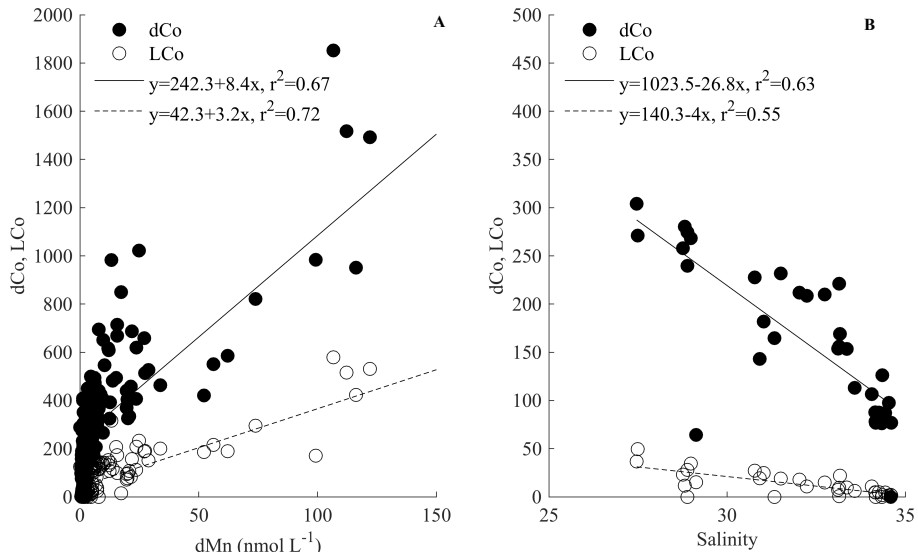



**Figure 12.**

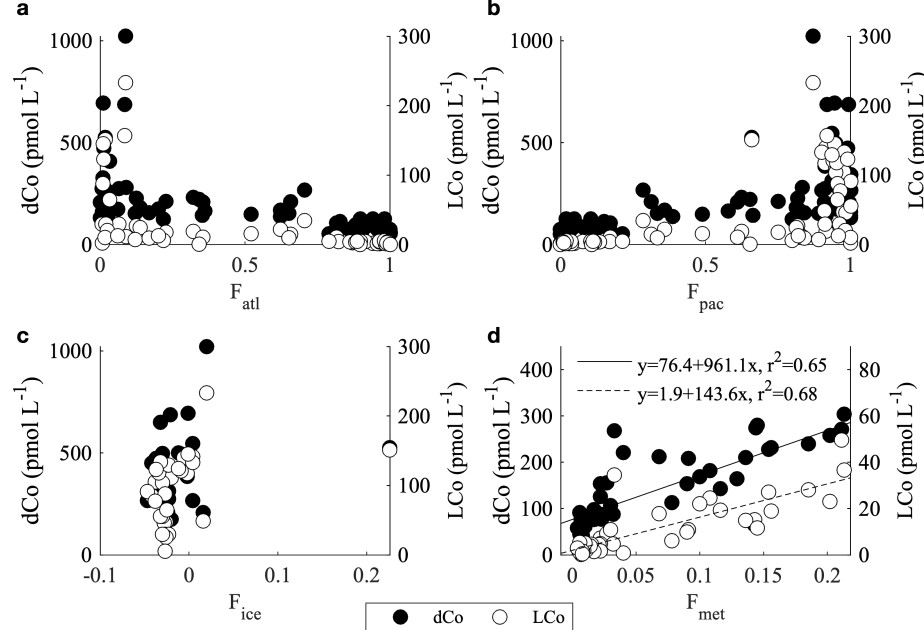



**Figure 13.**

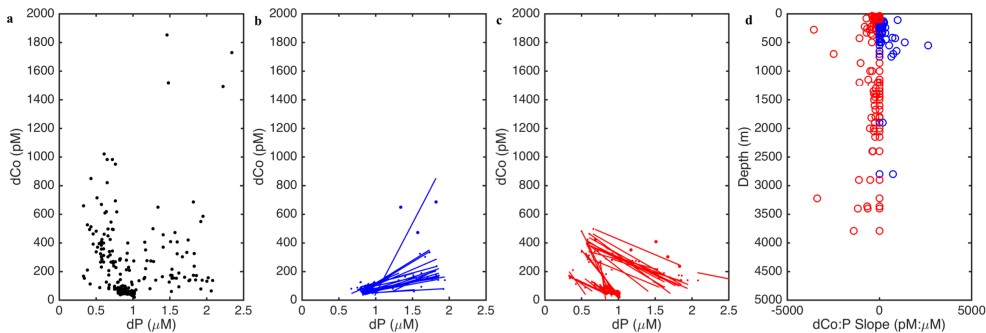



**Figure 14.**

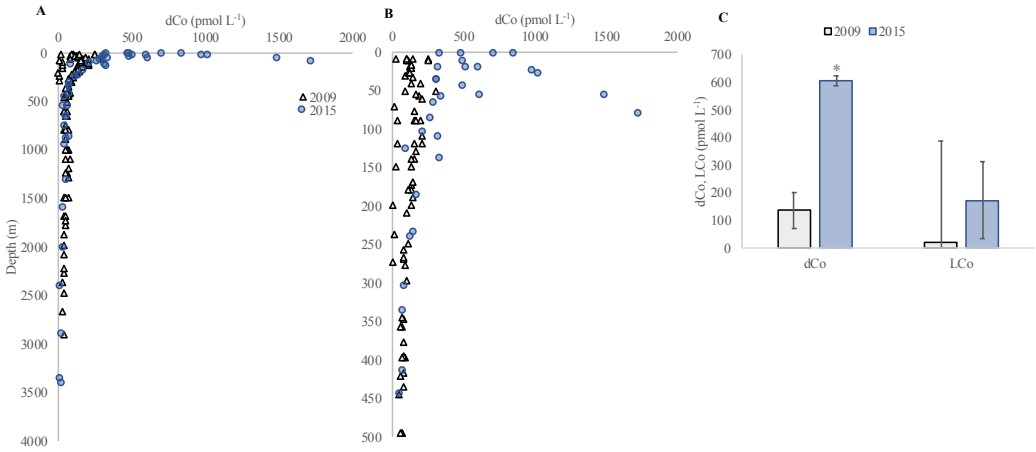



**Figure 15.**

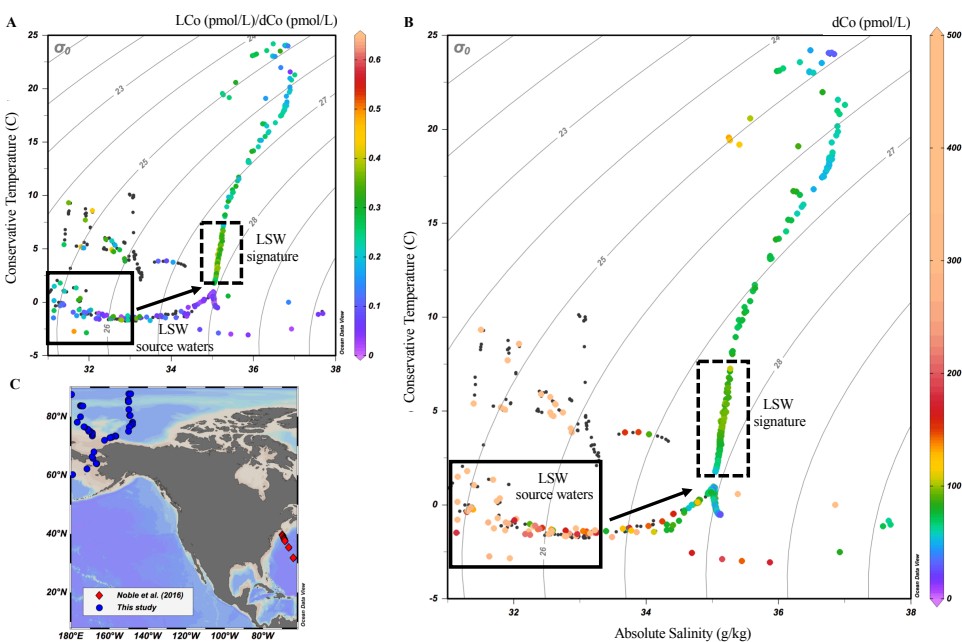





**Figure 16.**

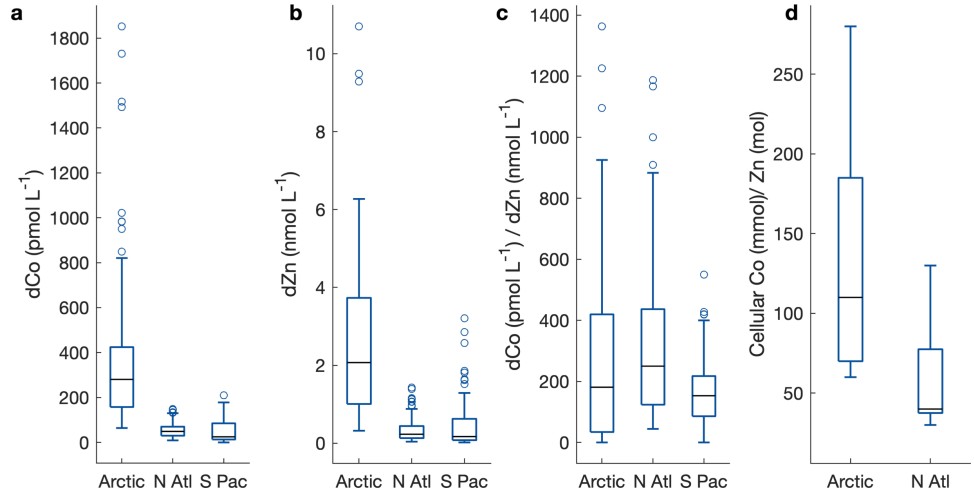





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
