# Peer review of "Elevated sources of cobalt in the Arctic Ocean"

_Biogeosciences, 2020_

## Referee Comment (RC1) · Anonymous Referee #1 · 16 Apr 2020

This is a well-written manuscript covering the cycling of cobalt in the Arctic Ocean. The cobalt measurements in the manuscript are high quality. The use of the Tagliabue's biogeochemical model with the addition of cobalt with the PISCES-v2 module provides insight into the likely sources of cobalt to the Aracti water column. I have not substantive comments on the manuscript.

Some minor comments -

line 269: the GitHub link doesn't contain any information

lines 369 to 375: this information is also presented in table 1 so it is repetitive - I suggest it is removed from here and just refer to the Table.

line 1060: Saito et al. 2001 reference is cited twice.

---

## Author Comment (AC1) · 3 May 2020

This is a well-written manuscript covering the cycling of cobalt in the Arctic Ocean. The cobalt measurements in the manuscript are high quality. The use of the Tagliabue's biogeochemical model with the addition of cobalt with the PISCES-v2 module provides insight into the likely sources of cobalt to the Arctic water column. I have not substantive comments on the manuscript.

Some minor comments -
line 269: the GitHub link doesn't contain any information

*Thank you, this has been fixed.*

lines 369 to 375: this information is also presented in table 1 so it is repetitive - I suggest it is removed from here and just refer to the Table.

*We have now shortened this section and combined it with the following section.*

line 1060: Saito et al. 2001 reference is cited twice.

*Thank you, this has now been fixed.*

---

## Referee Comment (RC2) · Anonymous Referee #2 · 14 May 2020

This is an important, well written manuscript. I think it should be published in Biogeosciences, however I have a number of comments that should be addressed first. My comments are listed in the order I came across them in the manuscript, not by importance.

Line 2: Has Co actually every been shown to limit phytoplankton growth in the ocean? Certainly not in many regions? Suggest to modify phrasing accordingly.

Line 40: Again, might be misleading to state that Co has been found to be the growth limiting nutrient?

Line 130: "pressurized filtered air' – N2 gas?

Line 131: States collection methods for Co, then nutrients, and then back to Co again.

[Figure]

Move nutrient sampling to end of paragraph?

Section 2.2: Not clear if samples for Co were acidified for storage or not?

Section 2.8:

- Briefly comment here on how PISCES-v2 performs in terms of physics/ice/rivers/macronutrients/chlorophyll in the Arctic Ocean, as these are key for interpreting the results. - Please state whether Co concentrations regulate phytoplankton growth in the model. - Please indicate whether there are Co binding ligands in the model, what their sources are etc. (as DOM complexion of dCo is inferred as important mechanism protecting against scavenging in the observational data, and could be an important factor contributing to differences with the model)

Lines 355–359: Repetition of how PHW can be identified?

Line 513–516: Perhaps rephrase to make clear that it is dissolution of Mn-oxides with Co bound to it is a Co source (not the Mn-oxide itself)

Section 3.5: There is a lot of discussion in this results section. The discussion including comparison to other regions and hypothesized mechanisms controlling Co distributions should be moved to the discussion section.

Line 553: Insert 'as indicated by dMn concentrations' when refereeing to shelf inputs?

Line 557: Replace 'diminished' by 'low' (i.e. diminished relative to what?)

Line 559: Which transect is being referred to here?

Lines 620–623: Sentence unclear – please rephrase

Line 679: Rephrase to state that it is a combination of restricted upper ocean scavenging in combination with continued deep water scavenging alongside restricted water mass mixing? i.e. Need to keep clear that there is expected lower scavenging in surface waters as these are agued to be important for leading to the high dCo there? If

this is not what the author's intended to say then this bit needs a little rephrasing.

Line 706: Rephrase to state that it is the observed increase in dCo over two time points. i.e., seasonal or inter-annual variability could explain this.

Paragraph starting line 736: Can this not be investigated with the PISCES model output? If the model is doing a good job in replicating Co in the Arctic for the correct mechanisms, then would export into the Atlantic ocean not be replicated by the model? If so then the authors can be a bit more quantitative about this statement (e.g. providing an approximate fraction of Arctic-sourced Co in the Atlantic). If the model is not replicating this, then this would still be interesting to comment upon (i.e. either the model or the proposed mechanism is incorrect).

Line 799: Again the authors should stress here that this is just two time points of observations and therefore not enough data to say whether Co concentrations in the Arctic are increasing with time.

Figures 8 and 9: Possible to make the colour bar numbers larger?

---

## Referee Comment (RC3) · Anonymous Referee #3 · 27 May 2020

The authors present novel data from an understudied region. The data appears of high quality, is definitely very interesting and as such should be published. Generally the paper is well written, but the methods and results section seem quite long. I like that the methods are detailed, but given that most methods have been described before I wonder if all details are required here and if the methods section could be condensed. The results section already contains quite some discussion. This should either be moved to the discussion, or perhaps a combined results/discussion section is more suitable for this paper (i.e. where the actual results section only briefly describes the basin wide distribution). The number of figures is also quite large and I encourage the authors to reconsider if they need them all as the sheer volume of data presented is a

bit overwhelming.

Generally, the discussion is interesting, but I thought it was strange that work already published from this cruise (e.g. Jensen et al., 2019) or the German/Dutch GEO-TRACES cruises (both 2015 and 2007 GEOTRACES IPY) is barely (or not at all) discussed. There is already quite some work that provides very nice context for the current study, e.g. the work on Fe and Fe binding ligands in the TPD from the GN04 transect, the Jensen Zn study or the Mn and Fe work from the GIPY 11 cruise (data in IDP). The same can be said for the comparison with the Atlantic where the comparison is made with data quite far south along the zonal GA03 rather than the also available meridional GA02 data that seems more relevant for the discussion of advection into the Atlantic.

In the section on correlations, mixing as a driving factor in such correlations is ignored (see also specific comments) and this is something that in my opinion should be addressed. The comparison with the data from 2009 is definitely interesting, but way overstated. Most importantly, 2 data points in time are not proof of a trend. Moreover, there could be seasonal variation (as it appears the stations were not occupied at the same time/season) and the stations locations are actually quite far apart. Notably the width (area) of the adjacent/closest shelf is very different and stations were positioned ifferently with respect to fluvial input and the Bering Strait. In this study it is argued that the shelf is the most important source of Co. Thus higher concentrations in the vicinity of the large shelf area of the Chukchi Sea and Bering Strait compared to the narrow Canada basin shelf are maybe not that surprising and this needs to be explored in context of the local hydrography and currents. I also noted (based on Fig 9) that the role that bacteria play differs between the regions for the 2009 and 2015 data.

The conclusions were not completely appropriate for the current ms. The data/ms did not show at all that the Co distribution has implications for Arctic ecosystems and it is unclear how the observation of a unique Co distribution affects future changes in micro nutrients. As stated above, I do not believe that Co was shown in this work to be

increasing over time. The idea that (changing) conditions in the Arctic affect the North Atlantic Ocean downstream is not new and this should be acknowledged.

Specific comments 40-43 limitation by cobalamin does not necessarily imply Co limitation as cobalamin production can be low regardless of the Co levels. So most of the cited studies do not demonstrate Co limitation.

76-80 it is stated there are regionally specific features, but the examples are not really specific regional features. Perhaps rephrase?

92 awkward sentence, please rephrase

94 what is meant with 'interpreting the role of external sources and internal cycling to the distribution'?

109 was sampled

131 can you compare filtered and unfiltered samples? Perhaps state this will be addressed later in the ms

295 this detection limit is at least an order of magnitude too high for open ocean Mn, notably in the deep. Is it a typo?

324 Fe was already defined

389 given that the data from the Canadian geotraces cruise was unfiltered, I do not think it is appropriate to call is dissolved (dCo)

408 how is the % sea ice melt determined?

427what is the % Pacific water based on?

427 here and elsewhere, the number of significant figures for Co concentrations does not seem to match the reported precision.

470 awkward sentence, please rephrase

471 what does 'that' refer too?

508 confused, 'capture the major processes contributing to modeled sources and sinks' not sure what is meant here.

516-517 Jensen et al 2019 argued that low oxygen in the sediments plays an important role for Zn and evidence for denitrification in the sediments was presented. This should also affect Co despite the fact that oxygen is not low in the water column. If denitrification occurs in the sediments, isn't it likely that also reductive dissolution of sedimentary Mn-oxides occurs? (however this discussion seems out of place in the results section)

516-534 this section is not as clearly written as the rest of the ms (specifically the last sentence was impossible to follow for me). Perhaps this can be remedied?

540-554 There is a problem with this section as for the Arctic (but also elsewhere, e.g. Aguilar-Islas A. and Bruland K. W. (2006)) it has been demonstrated that Mn in the surface of the open ocean basin is mainly derived from fluvial input, not sediments (Middag et al., 2011 (doi:10.1016/j.gca.2011.02.011). The latter study was not the exact same region, but fluvial input will be a strong source of Mn in this region too, and this needs to be discussed. However, Zn (Jensen et al., 2019) has been shown to have an important sedimentary source and might be a better proxy?

571 discussion of the recent TPD paper here seems appropriate (https://doi.org/10.1029/2019JC015920) as well as some other recent Fe work (https://dx.doi.org/10.3389/fmars.2018.00088; https://doi.org/10.1016/j.marchem.2017.10.005; https://doi.org/10.1029/2018JC014576 )

572/573 'track shelf inputs due to interactions between the sediment-water exchange processes' quite vague, not sure what this means/implies

582-584 What about deposition of riverine Co in the shelf sediments and subsequent remobilization?

590 also argued for Fe (https://doi.org/10.1016/j.marchem.2017.10.005; https://doi.org/10.1029/2018JC014576)

597 see mentioned refs for humic-like substances and Fe in the TPD, seems very relevant here.

662 what is meant with depth here? I assume the slopes are determined per station and the depth is the station depth or am I wrong? Please clarify

654-670 there is a growing body of work demonstrating mixing and water mass circulation is a primary factor in driving the slopes of metal-nutrient relationships that is ignored here while the mixing of Pacific and Atlantic origin water could have a strong effect (e.g. Vance et al., 2017, doi: 10.1038/ngeo2890; de Souza et al., 2018, doi:10.1016/j.epsl.2018.03.050; Middag et al., 2018, doi: 10.1016/j.epsl.2018.03.046;Weber et al., 2018, doi: 10.1126/science.aap8532; Middag et al., 2019, doi:10.1029/2018GB006034; Middag et al., 2020 doi: 10.3389/fmars.2020.00105).

676 continues?

702 after a 40% correction, the 2015 data is 400% higher than the 2009 data. This seems to be in contrast to line 688 where it is stated that without correction the 2015 data is 3.5 times higher.

703-704 could there be a factor of seasonality (did the sampling occur in same time of year relative to start of ice melt and river discharge)? And 2 data points in time hardly makes a trend! The difference is interesting for sure, but currently the significance is really overstated, as there is no way of telling what the Co concentrations were in other years. What about the enormous difference in the size of the nearby shelf regions between the 2 expeditions?

719-720 an increase in fluvial discharge as well as timing of ice melt could also affect primary productivity on the shelf and thus sedimentary oxygen conditions and Co supply from the sediments (similar to Zn; Jensen et al 2019). And what about increased SGD, could that play a role?

725 I have some issues with this section. First, it is very odd to compare only to the zonal Noble et al. study when in this discussion the comparison to the meridional Dulaquais study would make much more sense as that also has observations much closer to the Arctic (and also states: 'the LSW was characterized by relatively high DCo concentrations'). This data is available from the IDP. Moreover, LSW is not the only water mass of Arctic origin, also the deeper components of NADW are of Arctic origin (Denmark Strait Overflow Water and Iceland-Scotland Overflow Water). So if LSW is elevated in Co due to its Arctic origin, why is LSW elevated relative to ISOW and DSOW that are also of Arctic origin? This needs to be addressed.

749 not sure what the T-S plot shows/adds or how it supports the hypothesis; basically it shows that dCo is lower in LSW than in the source waters, but you do not need a T-S plot to show this.

773 where does the Zn data come from? According to the caption it is from this study, but this is the first mention of it. Again the comparison to the GA03 section rather than the more relevant GA02 section is very odd in my opinion as all data is accessible in the IDP and provides data (and insight from the associated publications) much closer to the Arctic. I really urge the authors to make use of the data (and insights) available from the international GEOTRACES efforts.

779 quite similar. What is this statement based on given that the medians are more than a factor of 2 apart? I see there is considerable overlap, but not sure if 'quite similar' is the observation all readers would make based on the presented graph. Some explanation seems required.

781 Bit of a jump from Co to total metal concentrations. For metals with different biogeochemistry this might be different and an increase in fluvial supply in the Arctic (of e.g. scavenged Al) might have no consequence for transport to the Atlantic.

783-785 do not follow this sentence; the total inventory of Zn is small compared to Zn? And why is the Jensen et al., 2019 only briefly mentioned here? As indicated above, the comparison to the cycling of Zn would have been relevant elsewhere in this ms too.

791-792 This ms has not demonstrated there is any influence of the Co distribution (or the changing Co concentrations) on the Arctic ecosystem, just that Co concentrations could be changing. Moreover, given that Co concentrations are high, I fail to see how a further increase in Co is affecting the ecosystem. And how does the unique Co distribution affect future changes in micro nutrients?

799 as stated before, this cannot be stated like this based on 2 data points in time!

805 similar interpretations were also invoked based on e.g. the micronutrient distributions along the GA02 section (e.g. Cd, Zn, Ni, Fe and Fe binding ligands, Co). I do not mind this is not a completely novel finding, but it is appropriate to acknowledge this idea was postulated before and in fact could strengthen the case for this study on Co.

Not all figures have units on the axis (color bar fig 8, y axis fig 11)

The cited references in the text are not all in reference list

---

## Author Comment (AC2) · 1 Jul 2020

**Review #2**

This is an important, well written manuscript. I think it should be published in Bio- geosciences, however I have a number of comments that should be addressed first. My comments are listed in the order I came across them in the manuscript, not by importance.

Line 2: Has Co actually every been shown to limit phytoplankton growth in the ocean? Certainly not in many regions? Suggest to modify phrasing accordingly.

*We clarified this statement to refer to cobalt's role as a co-factor in cyanocobalamin, which has been shown to limit phytoplankton growth in several regions of the oceans. Cobalt has also been shown to be serially limiting with nitrogen and iron (Browning et al., 2017). We updated this sentence to read, "Cobalt (Co) is an important bioactive trace metal that is the metal co-factor in cobalamin (vitamin $B_{12}$) which can limit or co-limit phytoplankton growth in many regions of the ocean."*

Line 40: Again, might be misleading to state that Co has been found to be the growth limiting nutrient?

*We amended this section to read, "Due to its low concentrations, strong organic complexation, and its presence in cobalamin, dCo or cobalamin have been found to be limiting or co-limiting nutrients for phytoplankton growth in several regions (Bertrand et al., 2007, 2015; Browning et al., 2017; Martin et al., 1989; Moore et al., 2013; Saito et al., 2005). Growth limitation can be due to either a lack of dCo, or cobalamin (Bertrand et al., 2012; Bertrand et al., 2007; Browning et al., 2017), as cobalamin is only synthesized by cyanobacteria and some archaea (Doxey et al., 2015). However, many phytoplankton utilize cobalamin for the synthesis of methionine (Yee and Morel, 1996; Zhang et al., 2009), and therefore must obtain it from the natural environment (Heal et al., 2017)."*

Line 130: "pressurized filtered air" – $N_2$ gas?

*Correct, the samples were collected using filtered pressurized air and not $N_2$ gas.*

Line 131: States collection methods for Co, then nutrients, and then back to Co again. Move nutrient sampling to end of paragraph?

*We have now moved the nutrient methods information to the end of the paragraph.*

Section 2.2: Not clear if samples for Co were acidified for storage or not?

*Samples were not acidified. We clarified this in Section 2.2: "...samples were kept refrigerated (4°C) and un-acidified until analysis (Hawco et al., 2016, 2018; Noble et al., 2016). LCo samples were double-bagged and stored at 4°C and un-acidified until analysis."*

Section 2.8:

- Briefly comment here on how PISCES-v2 performs in terms of physics/ice/rivers/macronutrients/chlorophyll in the Arctic Ocean, as these are key for interpreting the results.

*The PISCES model has been used extensively to examine global scale biogeochemical cycling. We did not choose here to make an extensive model evaluation study in the Arctic Ocean since this was not the focus of the paper. Instead, the goal was to explore how the model's cobalt sources, sinks and internal cycling performed and how they may help provide additional insight into the driving mechanisms behind the observed distributions. Thus we chose to focus the model evaluation on cobalt rather than a suite of extra tracers that would lengthen the manuscript and distract from its focus. More details on PISCES can be found in the cited reference publications (Aumont et al., 2015; 2017; Tagliabue et al., 2018).*

- Please state whether Co concentrations regulate phytoplankton growth in the model.

*Co concentrations do not regulate phytoplankton growth in the model. Uptake of Co by phytoplankton in the model is explicitly modeled based on a maximum cellular quota and allows for variable Co/C ratios (Tagliabue et al., 2018).*

- Please indicate whether there are Co binding ligands in the model, what their sources are etc. (as DOM complexion of dCo is inferred as important mechanism protecting against scavenging in the observational data, and could be an important factor contributing to differences with the model)

*Organic cobalt-binding ligands are modeled and are linked to the relative abundance of nanoplankton in the model as a means of representing the production of cobalamin or pseudocobalamin by picocyanobacteria communities. Co ligands also have a minimum deep ocean concentration in the model to stabilize dCo in the deep ocean and prevent scavenging. A few additional details about the model have been added to the manuscript in section 2.8 based on the suggestions above. Extensive details can be found in Tagliabue et al. (2018).*

Lines 355–359: Repetition of how PHW can be identified?

*We have shortened this to read, "The PHW can be clearly identified from the elevated macronutrient concentrations (Fig. 2D), and temperature maximum within the salinity range of 31-33 (Steele et al., 2004; Steele and Boyd, 1998) (Fig. 2A, C)."*

Line 513–516: Perhaps rephrase to make clear that it is dissolution of Mn-oxides with Co bound to it is a Co source (not the Mn-oxide itself).

*We have clarified this and rephrased.*

Section 3.5: There is a lot of discussion in this results section. The discussion including comparison to other regions and hypothesized mechanisms controlling Co distributions should be moved to the discussion section.

*We removed several lines of text from this section and combined it with the text in the second paragraph of section 4.1 of the discussion.*

Line 553: Insert 'as indicated by dMn concentrations' when refereeing to shelf inputs?

*This has been inserted.*

Line 557: Replace 'diminished' by 'low' (i.e. diminished relative to what?)

*This has been changed to indicate that there is diminished scavenging of dCo in the model relative to other ocean basins* (Tagliabue et al., 2018).

Line 559: Which transect is being referred to here?

*This has been changed to specify the GN01 transect.*

Lines 620–623: Sentence unclear – please rephrase

*This has been changed to read, "The interaction between rivers and shelves requires further study, as the shelf sediments might behave as "capacitor" for dCo, accumulating Co from rivers and sinking organic matter and then releasing Co to the overlying water during reductive dissolution in the sediments. Although the mechanism is uncertain, it is clear that the riverine source dominates the distribution observed near the North Pole where dCo and LCo concentrations remain high despite the distance from land, and that organic complexation likely plays a role in the distal transport of this dCo (Charette et al., 2020)."*

Line 679: Rephrase to state that it is a combination of restricted upper ocean scavenging in combination with continued deep water scavenging alongside restricted water mass mixing? i.e. Need to keep clear that there is expected lower scavenging in surface waters as these are argued to be important for leading to the high dCo there? If this is not what the author's intended to say then this bit needs a little rephrasing.

*This section has been amended as follows:*
*"This evidence, combined with the coinciding maxima observed in pCo and pMn, suggest that scavenging occurs in the upper water column, but that additional scavenging continues to occur in deeper waters. The elevated pCo concentrations in the deep Arctic compared to other regions (Lee et al., 2018) suggest that scavenging over long timescales continues to add to the pCo pool. The strong stratification in the Arctic likely prevents high concentrations of dCo from mixing between the modified surface waters, the PHW, and the deep Atlantic water (Steele et al., 2004). Thus, it is likely a combination of limited upper ocean scavenging, and strong stratification between water masses, that keeps the elevated dCo and LCo confined to the surface waters in Arctic, yielding the intense scavenged-like profile of Co in this region compared to other basins (Fig. 3)."*

Line 706: Rephrase to state that it is the observed increase in dCo over two time points. i.e., seasonal or inter-annual variability could explain this.

*We have updated this sentence to read, "While there is not enough data to state whether the river dCo flux has in fact changed over time in the Arctic and the observed changes could be due to seasonal or interannual variability, several other studies have documented an increase in river discharge due to increases in permafrost melt over time (Doxaran et al., 2015; Drake et al., 2018; Kipp et al., 2018; van der Loeff et al., 2018; Tank et al., 2016; Toohey et al., 2016)."*

Paragraph starting line 736: Can this not be investigated with the PISCES model output? If the model is doing a good job in replicating Co in the Arctic for the correct mechanisms, then would export into the Atlantic Ocean not be replicated by the model? If so then the authors can be a bit more quantitative about this statement (e.g. providing an approximate fraction of Arctic-sourced Co in the Atlantic). If the model is not replicating this, then this would still be interesting to comment upon (i.e. either the model or the proposed mechanism is incorrect).

*The downstream impact of the Arctic on the Atlantic Co distribution can be seen in the results of Tagliabue et al. 2018, but were not focused on in that paper. The clearest way to quantify the influence of the Arctic on the North Atlantic dCo distribution would be to "turn off" the Arctic sources of dCo in the model and see how that impacts distributions in the North Atlantic. However, this is not straightforward and fully masking all of the sources of Co in the Arctic is currently not possible in the current model framework. Alternative tests, such as picking experiments that led to strong reductions in Arctic Co levels (e.g. the experiments where bacterial activity did not affect the Co scavenging rate from Tagliabue et al, 2018) are also not ideal as the Arctic signal itself is not isolated.*

*We also felt we were not able to be quantitative about the Co source to the North Atlantic from observations alone because the transformations and source regions of Labrador Sea water are not entirely understood at this time (Le Bras et al., 2017), so it is difficult to say with the current data what proportional of the high dCo signal seen in Noble et al. (2016) is due to additional dCo sources on the western margin of the US.*

Line 799: Again the authors should stress here that this is just two time points of observations and therefore not enough data to say whether Co concentrations in the Arctic are increasing with time.

*We amended this section to make sure it is clear that the dataset is limited. It now reads, "Co was also shown in this work to be increasing over time on the shelf in the Canadian Arctic, possibly due to increases in river inputs from thawing permafrost, though this is difficult to constrain in the present limited dataset. Given the potential increase in Co over time in the Arctic and the modification of low-salinity Arctic waters as they exit the Arctic into the North Atlantic and the Labrador Sea, it is difficult to determine if there is a net flux of Co out of the Arctic and into the North Atlantic, however evidence in this work suggests that the distinct Co waters of the Arctic likely impact downstream micronutrient concentrations. These impacts are likely to become increasingly important in the future, with increased warming and changes to Co sources in the Arctic basin."*

Figures 8 and 9: Possible to make the colour bar numbers larger?

*Yes, we made the labels on Figures 8-10 larger.*

*References*

[revised manuscript text omitted]

---

## Author Comment (AC3) · 1 Jul 2020

**Review #3**

Review of 'Elevated sources of cobalt in the Arctic Ocean' by Bundy et al.

The authors present novel data from an understudied region. The data appears of high quality, is definitely very interesting and as such should be published. Generally the paper is well written, but the methods and results section seem quite long. I like that the methods are detailed, but given that most methods have been described before I wonder if all details are required here and if the methods section could be condensed. The results section already contains quite some discussion. This should either be moved to the discussion, or perhaps a combined results/discussion section is more suitable for this paper (i.e. where the actual results section only briefly describes the basin wide distribution). The number of figures is also quite large and I encourage the authors to reconsider if they need them all as the sheer volume of data presented is a bit overwhelming.

*Thank you for the comments. We have shortened the methods section where possible, and have moved some sentences from the results section to the discussion, particularly from the modeling results section (section 3.5). We realize the volume of data and the numbers of figures is quite large, however for completeness we have kept all of the figures and tables.*

Generally, the discussion is interesting, but I thought it was strange that work already published from this cruise (e.g. Jensen et al., 2019) or the German/Dutch GEOTRACES cruises (both 2015 and 2007 GEOTRACES IPY) is barely (or not at all) discussed. There is already quite some work that provides very nice context for the current study, e.g. the work on Fe and Fe binding ligands in the TPD from the GN04 transect, the Jensen Zn study or the Mn and Fe work from the GIPY 11 cruise (data in IDP). The same can be said for the comparison with the Atlantic where the comparison is made with data quite far south along the zonal GA03 rather than the also available meridional GA02 data that seems more relevant for the discussion of advection into the Atlantic.

*As discussed in the specific comments below, we have added many of these references throughout the discussion section. We have kept our current comparison to GA03, but we have mentioned that a similar signal was observed in GA02 in LSW.*

In the section on correlations, mixing as a driving factor in such correlations is ignored (see also specific comments) and this is something that in my opinion should be addressed. The comparison with the data from 2009 is definitely interesting, but way overstated. Most importantly, 2 data points in time are not proof of a trend. Moreover, there could be seasonal variation (as it appears the stations were not occupied at the same time/season) and the stations locations are actually quite far apart. Notably the width (area) of the adjacent/closest shelf is very different and stations were positioned differently with respect to fluvial input and the Bering Strait. In this study it is argued that the shelf is the most important source of Co. Thus higher concentrations in the vicinity of the large shelf area of the Chukchi Sea and Bering Strait compared to the narrow Canada basin shelf are maybe not that surprising and this needs to be

explored in context of the local hydrography and currents. I also noted (based on Fig 9) that the role that bacteria play differs between the regions for the 2009 and 2015 data.

*We discussed some of these suggestions below in the specific comments. We believe that the stratification in the Arctic basin which impedes mixing between the Pacific waters and Atlantic waters as shown in this work (and others), would suggest that the impact of mixing of water masses on the Co:P stoichiometry observed is minimal. The water masses in this region have very little exchange (Figure 2) and thus the primary drivers of changes in the deep Co:P ratios are likely due to internal cycling.*

*We have also noted this extensively below, but we did not intend to suggest that our dCo from 2009 and 2015 is definitive evidence that Co is increasing over time in the Arctic, as we discussed many caveats in the manuscript. However, we do feel it is an important observation to document in our work.*

The conclusions were not completely appropriate for the current ms. The data/ms did not show at all that the Co distribution has implications for Arctic ecosystems and it is unclear how the observation of a unique Co distribution affects future changes in micro nutrients. As stated above, I do not believe that Co was shown in this work to be increasing over time. The idea that (changing) conditions in the Arctic affect the North Atlantic Ocean downstream is not new and this should be acknowledged.

*We have reworked section 4.3 and 4.4 of the discussion to make it clearer that we were not trying to say that our data shows that dCo has unequivocally been increasing over time in the Arctic. We have pointed out that our data suggests that dCo is increasing, however we recognize that the data is limited. However, since others have noted the same trend in other tracers, we do not believe our conclusions our unjustified. We have cited several papers that have also noted increases in shelf-derived tracers in the Arctic over time and their affects to the downstream North Atlantic (Charette et al., 2020; Kipp et al., 2018; van der Loeff et al., 2018).*

Specific comments 40-43 limitation by cobalamin does not necessarily imply Co limitation as cobalamin production can be low regardless of the Co levels. So most of the cited studies do not demonstrate Co limitation.

*We clarified this sentence to read, "Due to its low concentrations, strong organic complexation, and its presence in cobalamin, dCo or cobalamin have been found to be limiting or co-limiting nutrients for phytoplankton growth in several regions (Bertrand et al., 2007, 2015; Browning et al., 2017; Martin et al., 1989; Moore et al., 2013; Saito et al., 2005). Growth limitation can be due to either a lack of dCo, or cobalamin (Bertrand et al., 2012; Bertrand et al., 2007; Browning et al., 2017), as cobalamin is only synthesized by cyanobacteria and some archaea (Doxey et al., 2015)."*

76-80 it is stated there are regionally specific features, but the examples are not really specific regional features. Perhaps rephrase?

*This sentence was removed.*

92 awkward sentence, please rephrase

*This has been rephrased as "This study examined dCo, LCo, and pCo in two different transects in the Canadian sector of the Arctic Ocean."*

94 what is meant with 'interpreting the role of external sources and internal cycling to the distribution'?

*We meant that we used the model to evaluate our hypotheses about the key factors in controlling dCo distributions in the Arctic. We changed this sentence to, "We then used a Co biogeochemical model (Tagliabue et al., 2018) in order to evaluate hypotheses about the role of external sources and internal cycling to the observed Co distributions, the potential of the Arctic to be a net source of Co to the North Atlantic, and to identify Co sources and sinks that may be sensitive to future changes in this rapidly changing ocean basin."*

109 was sampled

*We left this sentence as is.*

131 can you compare filtered and unfiltered samples? Perhaps state this will be addressed later in the ms

*This was addressed later in the manuscript.*

295 this detection limit is at least an order of magnitude too high for open ocean Mn, notably in the deep. Is it a typo?

*This is not a typo, and refers to the detection limit of shipboard flow injection analyses of dMn and not ICP-MS analyses.*

324 Fe was already defined

*Thank you, this has been fixed.*

389 given that the data from the Canadian geotraces cruise was unfiltered, I do not think it is appropriate to call is dissolved (dCo)

*This section is about the GN01 data, which are all filtered.*

408 how is the % sea ice melt determined?

*This is determined from $\delta^{18}O$ data (Newton et al., 2013). This reference has been added.*

427what is the % Pacific water based on?

*Same as above.*

427 here and elsewhere, the number of significant figures for Co concentrations does not seem to match the reported precision.

*All data presented have the correct number of significant figures.*

470 awkward sentence, please rephrase

*This has been rephrased.*

471 what does 'that' refer too?

*This has been rephrased to, "Similar to dCo, there was no observable enhancement of LCo in PHW, with LCo distributions closely following that of dCo and other shelf-enhanced trace metals such as dFe and dMn."*

508 confused, 'capture the major processes contributing to modeled sources and sinks' not sure what is meant here.

*This has been amended to, "In order to explore the major processes contributing to the modeled dCo sources and sinks, the proportion of the dCo signal in two distinct depth horizons was further investigated using a set of sensitivity experiments."*

516-517 Jensen et al 2019 argued that low oxygen in the sediments plays an important role for Zn and evidence for denitrification in the sediments was presented. This should also affect Co despite the fact that oxygen is not low in the water column. If denitrification occurs in the sediments, isn't it likely that also reductive dissolution of sedimentary Mn-oxides occurs? (however this discussion seems out of place in the results section)

*Yes, it is possible that there may be denitrification occurring in the sediments which could impact the dMn and dCo distributions. This is accounted for indirectly in the model by the sediment Co source being a function of the particulate organic carbon (POC) flux, which is a primary driver of anoxic sediments and thus denitrification.*

516-534 this section is not as clearly written as the rest of the ms (specifically the last sentence was impossible to follow for me). Perhaps this can be remedied?

*We have re-worded several sentences in this section.*

540-554 There is a problem with this section as for the Arctic (but also elsewhere, e.g. Aguilar-Islas A. and Bruland K. W. (2006)) it has been demonstrated that Mn in the surface of the open ocean basin is mainly derived from fluvial input, not sediments (Middag et al., 2011 (doi:10.1016/j.gca.2011.02.011). The latter study was not the exact same region, but fluvial input will be a strong source of Mn in this region too, and this needs to be discussed. However, Zn (Jensen et al., 2019) has been shown to have an important sedimentary source and might be a better proxy?

*Middag et al. (2011) and Charette et al. (2020) both suggest that dMn in the Arctic has both a fluvial and sedimentary source. In the Arctic, it is difficult to disentangle the shelf and riverine processes, as the riverine inputs interact with the shelf before being transported to the open basins (Kipp et al. 2018). The same is true for the dCo, which we mention in section 4.1. We have amended this section to highlight that although we believe the shelf signal to be the primary dCo source, we note that the fluvial inputs are very important in the open basin due to the TPD.*

571 discussion of the recent TPD paper here seems appropriate as well as some recent Fe work

*Yes, these references have now been updated since the recent publication of Charette et al. (2020), Colombo et al., (2020) and Tonnard et al. (2020).*

572/573 'track shelf inputs due to interactions between the sediment-water exchange processes' quite vague, not sure what this means/implies

*We have clarified this sentence to indicate that radium is a tracer for shelf inputs.*

582-584 What about deposition of riverine Co in the shelf sediments and subsequent remobilization?

*Yes, this could be another process on the shelf that is contributes to elevated dCo and is mentioned later on in this section.*

590 also argued for Fe (https://doi.org/10.1016/j.marchem.2017.10.005; https://doi.org/10.1029/2018JC014576)

*We have added these references.*

597 see mentioned refs for humic-like substances and Fe in the TPD, seems very relevant here.

*Yes, these have been added.*

662 what is meant with depth here? I assume the slopes are determined per station and the depth is the station depth or am I wrong? Please clarify

*We have reworded this to be "versus depth."*

654-670 there is a growing body of work demonstrating mixing and water mass circulation is a primary factor in driving the slopes of metal-nutrient relationships that is ignored here while the mixing of Pacific and Atlantic origin water could have a strong effect (e.g. Vance et al., 2017, doi: 10.1038/ngeo2890; de Souza et al., 2018, doi:10.1016/j.epsl.2018.03.050; Middag et al., 2018, doi: 10.1016/j.epsl.2018.03.046;Weber et al., 2018, doi: 10.1126/science.aap8532; Middag et al., 2019, doi:10.1029/2018GB006034; Middag et al., 2020 doi: 10.3389/fmars.2020.00105).

*This is very important to consider in other ocean basins, but there is very little mixing in the Arctic between water masses due to the strong stratification, so we do not think this is significant*

*here. This is also likely not as important for dCo, which has a much shorter residence time (~ 200 years) compared to some of the longer residence time elements mentioned in these references.*

676 continues?

*This has been changed.*

702 after a 40% correction, the 2015 data is 400% higher than the 2009 data. This seems to be in contrast to line 688 where it is stated that without correction the 2015 data is 3.5 times higher.

*This has been corrected.*

703-704 could there be a factor of seasonality (did the sampling occur in same time of year relative to start of ice melt and river discharge)? And 2 data points in time hardly makes a trend! The difference is interesting for sure, but currently the significance is really overstated, as there is no way of telling what the Co concentrations were in other years. What about the enormous difference in the size of the nearby shelf regions between the 2 expeditions?

*Yes, both samplings were done in the same month of the year (October). We have discussed many of these points extensively in this section, and have been transparent about the caveats. Many others have also observed increases in fluvial and shelf tracers over time in the Arctic however (Doxaran et al., 2015; Drake et al., 2018; Kipp et al., 2018; van der Loeff et al., 2018; Tank et al., 2016; Toohey et al., 2016), so we do believe that our data could be pointing to an increase in dCo over time as well. We have thoroughly explained these caveats in this section.*

*"The increase in dCo over time in the Arctic is interesting, and has been documented for other tracers in the Arctic. Kipp et al. (2018) and van der Loeff et al. (2018) noted that $^{228}$Ra has increased over time in the central Arctic. They suggest that increases in shelf and/or river inputs from thawing permafrost are the source of this elevated $^{228}$Ra (Kipp et al., 2018; van der Loeff et al., 2018). The increase in metal inventories over time on Arctic shelves is consistent with this observation. The majority of the variance (~70%) in dCo in the upper 100 m on the U.S. GEOTRACES transect could be explained by a shelf source, and the remainder was likely associated with river inputs (Fig. 11). If these sources are similar to the sources of dCo in 2009, then an increase in either a shelf or river flux could be responsible for the dramatic increase in dCo over time. While there is not enough data to state whether the river dCo flux has in fact changed over time in the Arctic and the observed changes could be due to seasonal or interannual variability, several other studies have documented an increase in river discharge due to increases in permafrost melt over time (Doxaran et al., 2015; Drake et al., 2018; Kipp et al., 2018; van der Loeff et al., 2018; Tank et al., 2016; Toohey et al., 2016). The increase in river discharge has the potential to considerably increase trace metal inventories in the future Arctic Ocean, perhaps particularly for those metals that are strongly organically complexed, thus protecting against scavenging in the estuarine mixing zone (Bundy et al., 2015). These increases in metals over time will have implications for metal stoichiometries and phytoplankton growth in a changing Arctic Ocean."*

719-720 an increase in fluvial discharge as well as timing of ice melt could also affect primary productivity on the shelf and thus sedimentary oxygen conditions and Co supply from the sediments (similar to Zn; Jensen et al 2019). And what about increased SGD, could that play a role?

*We do not think that SGD could be playing a role here because there are no marine terminating glaciers in this region to our knowledge. Primary production certainly could play a role, and these have been mentioned in this section.*

725 I have some issues with this section. First, it is very odd to compare only to the zonal Noble et al. study when in this discussion the comparison to the meridional Dulaquais study would make much more sense as that also has observations much closer to the Arctic (and also states: 'the LSW was characterized by relatively high DCo concentrations'). This data is available from the IDP. Moreover, LSW is not the only water mass of Arctic origin, also the deeper components of NADW are of Arctic origin (Denmark Strait Overflow Water and Iceland-Scotland Overflow Water). So if LSW is elevated in Co due to its Arctic origin, why is LSW elevated relative to ISOW and DSOW that are also of Arctic origin? This needs to be addressed.

*We have added the Dulaquais et al. (2014) reference in this section. We have discussed in this section that the LSW signature is likely a combined signal of Arctic inputs and additional dCo inputs picked up on the shelf in the Labrador Sea, and that is part of the reason why we do not think there is a similarly visible signal in ISOW and DSOW. Additionally, the high dCo is confined to the upper water column in the Arctic and thus is less likely to contribute to these deep water masses. LSW is also fresher and has lower silicate compared to ISOW and DSOW (Jenkins et al. 2015), additionally suggesting an influence from surface waters.*

749 not sure what the T-S plot shows/adds or how it supports the hypothesis; basically it shows that dCo is lower in LSW than in the source waters, but you do not need a T-S plot to show this.

*We have kept this figure because we think it is the best way to show the two datasets concurrently.*

773 where does the Zn data come from? According to the caption it is from this study, but this is the first mention of it. Again the comparison to the GA03 section rather than the more relevant GA02 section is very odd in my opinion as all data is accessible in the IDP and provides data (and insight from the associated publications) much closer to the Arctic. I really urge the authors to make use of the data (and insights) available from the international GEOTRACES efforts.

*The Zn data from the Arctic is from Jensen et al. (2019) and S. John (unpublished). The remaining data is from the GEOTRACES IDP 2017. Both Noble et al. (2016) and Dulaquais et al. (2014) observed similar signatures of high dCo in LSW, so we feel like either dataset is appropriate for this comparison. We have now discussed this more thoroughly in this section.*

779 quite similar. What is this statement based on given that the medians are more than a factor of 2 apart? I see there is considerable overlap, but not sure if 'quite similar' is the observation all readers would make based on the presented graph. Some explanation seems required.

*We have changed this to be "similar."*

781 Bit of a jump from Co to total metal concentrations. For metals with different biogeochemistry this might be different and an increase in fluvial supply in the Arctic (of e.g. scavenged Al) might have no consequence for transport to the Atlantic.

*We of course acknowledge that there will not be increases in all other trace metals, though it is plausible for those that show similar correlations with shelf and fluvial inputs (Charette et al. 2020).*

783-785 do not follow this sentence; the total inventory of Zn is small compared to Zn? And why is the Jensen et al., 2019 only briefly mentioned here? As indicated above, the comparison to the cycling of Zn would have been relevant elsewhere in this ms too.

*Here, we were stating that the total inventory of dCo in the ocean is much smaller than dZn, so small changes to dCo sources may have a disproportionate impact compared to increases in dZn fluxes. We have added some discussion of the Zn distributions throughout the manuscript, while being mindful of length.*

791-792 This ms has not demonstrated there is any influence of the Co distribution (or the changing Co concentrations) on the Arctic ecosystem, just that Co concentrations could be changing. Moreover, given that Co concentrations are high, I fail to see how a further increase in Co is affecting the ecosystem. And how does the unique Co distribution affect future changes in micro nutrients?

*This sentence was meant to highlight the distinct distributions of dCo in this basin compared to other open ocean regions (Figure 3). We also discussed how because the primary sources of dCo in this basin were found to be from a combination of shelf sediments and rivers, and that these sources have been shown to be increasing over time for many other tracers, that it is possible for dCo to continue to change over time as well.*

799 as stated before, this cannot be stated like this based on 2 data points in time!

*We have clarified throughout the manuscript that we are merely provide intriguing evidence that dCo is increasing over time in the Arctic. We have also added the following sentence in section 4.3, "We recognize these two Arctic dCo datasets are limited in temporal coverage and have methodological differences; however, we felt a responsibility to transparently present these observations of dCo increases in the Arctic Ocean to raise community awareness of this potential environmental change."*

805 similar interpretations were also invoked based on e.g. the micronutrient distributions along the GA02 section (e.g. Cd, Zn, Ni, Fe and Fe binding ligands, Co). I do not mind this is not a completely novel finding, but it is appropriate to acknowledge this idea was postulated before and in fact could strengthen the case for this study on Co.

*These other datasets have been mentioned in the preceding section.*

Not all figures have units on the axis (color bar fig 8, y axis fig 11) The cited references in the text are not all in reference list

*Both have been amended.*

**References**

[revised manuscript text omitted]

---

## Author Response (AR2)

The authors have done a good job on the revisions. However, I do not agree with their assessment that mixing can be ignored in the Co-P relationships. Even though stratification is strong, eventually the Pacific origin water does mix with the Atlantic origin water, otherwise we would be able to recognize Pacific water exiting through Fram Stait. Additionally, stratification is strong in summer but given that this is a region of deep water formation, very significant mixing occurs driven by winter cooling and brine rejection at other times of year. Maybe not directly along the transect, but it does strongly affect the water masses and circulation in the Arctic.

The fact that there are distinct water masses/layers going from surface to PHW to deep waters, will also affect the element profiles and thus slopes of an element-element relationship, even if there would be no mixing at all. For example, I suspect that 3 data points in surface water and 2 data point in PHW will give a different slope compared to a relationship comprising of 2 data points in PHW and 3 in underlying deeper water. It is not clear if this is accounted for in the assessment of slopes.

Response: We have added a sentence in section 4.2 to clarify that some of the negative dCo:P slopes could represent dilution of Arctic waters with deep Atlantic waters (similar to Dulaquais et al., 2014b):

"The negative slopes at the base of the profiles could also represent the dilution of dCo in the deep Arctic with lower dCo Atlantic water, as noted in the western Atlantic Ocean (Dulaquais et al., 2014b). However, it is unlikely that dilution alone accounts for the negative slopes observed throughout the water column."

However, in our dataset the negative slopes, and thus the apparent scavenging of dCo, is present throughout the entire water column and not just at the boundaries between the high dCo waters in the surface and the PHW in the upper mesopelagic. It is very unlikely that in this particular transect that mixing alone can account for the strong negative slopes observed.

Specific comments:

Line 165 did not see discussion on storage in 3.2.2.

Response: We have amended this to refer only to section 4.3.

(Irrelevant for this paper where the correlation between Co and Mn is assessed in the surface layer with high Mn concentrations, but have to say that the detection limit of 0.55 nM for Mn is not specific for a flow injection analysis, detection limits of an order of magnitude lower (down to <0.01 nM) have been demonstrated for Mn by flow injection)

*Response: The data used here is the shipboard dMn data and this was the reported detection limit for those analyses.*

I insist it should be explicitly stated here Mn also has a fluvial source in the Arctic (this is indirectly noted in the next paragraph, but it should be explicitly clear for others who might want to follow a similar approach)

Response: This has now also been noted in this paragraph, though the source of dMn observed from rivers along our transect was quite limited compared to the other trace metals (Charette et al., 2020):

"Mn is known to be an excellent tracer of sediment input due to the high solubility of reduced Mn from anoxic sediments (Johnson et al., 1992; März et al., 2011; McManus et al., 2012; Noble et al., 2012), though there was also a limited source of dMn from rivers in this region (Charette et al., 2020)."

**528 the variance explained and R2 in the associated figure do not match**

**Response: This has been fixed.**

4.3 I feel it should also be noted the sampling locations were quite different so spatial patterns could be important too. Notably given that the shelf is supposed to be the largest source, the dramatic variations in shelf width through the Arctic with a narrow shelf on the Canada basin side and wide shelf on the Chukchi side combined with inflow through the Bering Strait could play an important role too.

**Response: We have added a sentence on shelf-width:**

"Some of the samples from 2009 were also collected over a narrower region of the shelf compared to those in 2015, so shelf width could also be an important factor in the observed increase in dCo. Thus, although we cannot quantify with certainty the percent increase in dCo over time in the Canadian sector of the Arctic, it is possible that an increase in dCo was observed."

confusing, waters that pass through the Canadian archipelago exit via Fram strait?

Response: We are stating in this sentence that LSW contains a mixture of water that was modified both on Arctic shelves and on the Labrador Sea shelf prior to forming intermediate waters, so we cannot say for certain where the high dCo concentrations in these waters originates.

rephrase for clarity, now it is seemingly stated that the total Zn inventory is small compared to Zn

Response: This has been rephrased.

not shown but suggested

Response: This has been rephrased.

**Elevated sources of cobalt in the Arctic Ocean**

Randelle M. Bundy1,a,\*, Alessandro Tagliabue2, Nicholas J. Hawco1,4, Peter L. Morton3, Benjamin S. Twining4, Mariko Hatta5, Abigail E. Noble1,b, Mattias R. Cape1,a, Seth G. John6, Jay T. Cullen7 and Mak A. Saito1

 1Department of Marine Chemistry and Geochemistry, Woods Hole Oceanographic Institution, Woods Hole, MA, USA
 2School of Environmental Sciences, University of Liverpool, Liverpool, United Kingdom 3National High Magnetic Field Laboratory, Tallahassee, FL, USA
 4Bigelow Laboratory for Ocean Sciences, East Boothbay, ME, USA
 5Department of Oceanography, University of Hawai'i at Manoa, Honolulu, HI
 6Department of Earth Sciences, University of Southern California, Los Angeles, CA, USA
 7School of Earth and Ocean Sciences, University of Victoria, Victoria, BC, Canada
 a School of Oceanography, University of Washington, Seattle, WA, USA
 bCalifornia Department of Toxic Substances Control, Sacramento, CA, USA

\*corresponding author: msaito@whoi.edu

Keywords: cobalt, GEOTRACES, Arctic Ocean, biogeochemical model

Running header: Elevated cobalt in the Arctic

**1 Abstract**

Cobalt (Co) is an important bioactive trace metal that is the metal co-factor in cobalamin (vitamin B12) which can limit or co-limit phytoplankton growth in many regions of the ocean. 3 Total dissolved and labile Co measurements in the Canadian sector of the Arctic Ocean during 4 U.S. GEOTRACES Arctic expedition (GN01) and the Canadian International Polar Year-5 6 GEOTRACES expedition (GIPY14) revealed a dynamic biogeochemical cycle for Co in this 7 basin. The major sources of Co in the Arctic were from shelf regions and rivers, with only minimal contributions from other freshwater sources (sea ice, snow) and aeolian deposition. The 8 most striking feature was the extremely high concentrations of dissolved Co in the upper 100 m, 9 with concentrations routinely exceeding 800 pmol L-1 over the shelf regions. This plume of high 10 Co persisted throughout the Arctic basin and extended to the North Pole, where sources of Co 11 shifted from primarily shelf-derived to riverine, as freshwater from Arctic rivers was entrained in 12 13 the Transpolar Drift. Dissolved Co was also strongly organically-complexed in the Arctic, ranging from 70-100% complexed in the surface and deep ocean, respectively. Deep water 14 15 concentrations of dissolved Co were remarkably consistent throughout the basin (~55 pmol L-1), 16 with concentrations reflecting those of deep Atlantic water and deep ocean scavenging of dissolved Co. A biogeochemical model of Co cycling was used to support the hypothesis that the 17 majority of the high surface Co in the Arctic was emanating from the shelf. The model showed 18 19 that the high concentrations of Co observed were due to the large shelf area of the Arctic, as well 20 as dampened scavenging of Co by manganese (Mn)-oxidizing bacteria due to the lower temperatures. The majority of this scavenging appears to have occurred in the upper 200 m, with 21 22 minimal additional scavenging below this depth. These limited temporal results are consistent 23 with other tracers showing increased continental fluxes to the Arctic ocean, and imply both dCo and LCo increasing over time on the Arctic shelf. These elevated surface concentrations of Co 24 likely lead to a net flux of Co out of the Arctic, with implications for downstream biological 25 uptake of Co in the North Atlantic and elevated Co in North Atlantic Deep Water. Understanding 26 the current distributions of Co in the Arctic will be important for constraining changes to Co 27 inputs resulting from regional intensification of freshwater fluxes from ice and permafrost melt 28 29 in response to ongoing climate change.

**31 1. Introduction**

33 Cobalt (Co) is an essential micronutrient in the ocean. It is utilized by eukaryotic phytoplankton 34 as a substitute for zinc (Zn) in the metalloenzyme carbonic anhydrase (Lane and Morel, 2000; 35 Sunda and Huntsman, 1995; Yee and Morel, 1996), and cvanobacteria have an absolute 36 requirement for Co (Hawco and Saito, 2018; Saito et al., 2002; Sunda and Huntsman, 1995). Co 37 is also the metal center in the micronutrient cobalamin, or vitamin B12. In most ocean basins, dissolved Co (dCo;  $< 0.2 \mu m$ ) is extremely scarce in surface waters ( $< 10 \text{ 
[revised manuscript text omitted]